

# On the impact of canopy environmental variables on the diurnal dynamics of the leaf and canopy water and carbon dioxide exchange

Raquel González-Armas[1], Jordi Vilà-Guerau de Arellano[1,2], Mary Rose Mangan[1], Oscar Hartogensis[1], and Hugo de Boer[3]

[1]Meteorology and Air Quality Section, Wageningen University Research, 6708 PB Wageningen, the Netherlands
[2]Atmospheric Chemistry Department, Max Planck Institute for Chemistry, 55128 Mainz, Germany
[3]Department of Environmental Sciences, Faculty of Geosciences, Utrecht University, 3508 TA Utrecht, the Netherlands

**Correspondence:** Raquel González-Armas (raquel.gonzalezarmas@wur.nl)

**Abstract.** Quantifying water vapor and carbon dioxide exchange dynamics between land and atmosphere through observations and modelling is necessary to reproduce and project near surface climate in coupled land-atmosphere models. The exchange of water and carbon dioxide ($CO_2$) occurs at the leaf surfaces (leaf level) and in a net manner through the exchanges at all the leaf surfaces composing the vegetation canopy and at the soil surface (canopy level). These exchanges depend on the
meteorological forcings imposed by the overlying atmosphere (atmospheric boundary layer level). In this manuscript, we investigate the effect of four canopy environmental variables (photosynthetic active radiation (PAR), water vapor pressure deficit (VPD), air temperature (T) and atmospheric $CO_2$ concentration ($C_a$)) on the local individual leaf exchange and canopy exchange of water and $CO_2$ at hourly time scales and the effect of atmospheric boundary layer (ABL) processes on the local exchange.

To that end, we simultaneously investigated the exchanges of water and $CO_2$ at leaf level and canopy level for an alfalfa field in Northern Spain during a day in the summer of 2021. We used comprehensive observations ranging from stomatal conductance to ABL measurements collected during the Land Surface Interactions with the Atmosphere in the Iberian Semi-Arid Environment (LIAISE) experiment. To support the observational analysis, we used an integrative mixed-layer atmospheric model (CLASS) that have representations at all considered levels. To relate how temporal changes of the four environmental
variables modify the fluxes of water and $CO_2$, we studied tendency equations of the leaf gas exchange. These mathematical expressions quantify the temporal evolution of the leaf gas exchange as a function of the temporal evolution of PAR, VPD, T and $C_a$. To investigate the effects of ABL processes on the local exchange, we developed three modelling experiments that impose surface radiative perturbations by a cloud passage (which perturbed PAR, T and VPD), entrainment of dry air from the free troposphere (which perturbed VPD) and advection of cold air (which perturbed T and VPD).

Model results and observations matched the leaf gas exchange (with $r^2$ between 0.23 and 0.67) and canopy gas exchange (with $r^2$ between 0.90 and 0.95). The tendency equations of the modelled leaf gas exchange during the studied day revealed that the temporal dynamics of PAR were the main contributor to the temporal dynamics of the leaf gas exchange with atmospheric $CO_2$ temporal dynamics being the least important contributor. From the three modelling experiments with ABL perturbations, the surface radiative changes induced by a cloud perturbed the $CO_2$ exchange the most, whereas all of them perturbed the
water exchange to a similar extent. Second order effects on the dynamics of the leaf gas exchange were also identified using



the tendency equations. For instance, the decrease of net $CO_2$ assimilation rate during the cloud due to a decrease in surface radiation was further enhanced due to the decrease in air temperature also associated with the cloud. With this research we showcase that the proposed tendency equations can disentangle the effect of environmental variables on the leaf exchange of water and $CO_2$ with the atmosphere as represented in land-surface parameterization schemes and become a useful tool to

analyze these schemes in weather and climate models.

## 1 Introduction

The exchanges of water and carbon dioxide ($CO_2$) between land and atmosphere are essential components to constrain and understand the water and carbon cycles. Because of the complex dynamic interactions between soil, vegetation and atmosphere, the net surface fluxes of $CO_2$ and water vapor, known as net ecosystem exchange (NEE) and evapotranspiration (ET), remain

difficult to reproduce by current land surface models (LSMs). Intercomparison studies (Henderson-Sellers et al., 1995; Chen et al., 1997; Holtslag et al., 2013; Best et al., 2015; Restrepo-Coupe et al., 2017; Renner et al., 2021) have shown systematic deviations between observed and modelled ET and NEE. Additionally, they have shown discrepancies among the different LSMs considered. For instance, Renner et al. (2021) compared the estimation of heat surface fluxes of 13 different LSMs driven by observed meteorological conditions at 20 FLUXNET sites. When assessing the performance to reproduce heat fluxes, they

considered both the magnitude and a phase lag to incoming shortwave solar radiation. In their study they concluded that all LSMs showed a poor representation of the evaporative fraction and phase lag. The authors also highlighted the importance of systematic evaluations of the diurnal dynamics of the fluxes in order to improve the understanding and predictive capacity of the near-surface climate.

To perform a systematic evaluation of the diurnal dynamics of ET and NEE, multiple spatial scales, ranging from the size of

the stomata (10-100 $\mu$m) to the size of the atmospheric boundary layer ($\approx$ 1 km) must be considered. We have broadly divided the spatial scales into three discrete spatial levels: leaf level, canopy level and atmospheric boundary layer level (Fig. 1). The leaf and the canopy levels are two distinct levels where the exchange of water and $CO_2$ occur. They are different because at the leaf level our system is the leaf surface which experiences certain environmental conditions whereas at the canopy level the system comprises the whole vegetation canopy (with changing environmental conditions experienced by a leaf depending on

its location in the canopy) including the soil. The third level is the atmospheric boundary layer (ABL) level, which is the part of the atmosphere in immediate contact with the surface. The ABL reacts to the dynamics of the surface and imposes forcings to it.

At the leaf level the dynamic exchange of water and $CO_2$ is crucially influenced by in-canopy light, temperature, humidity and $CO_2$ concentration as well as the plant responses to these environmental conditions in terms of photosynthesis, transpiration

and stomatal conductance. Several models and theories have been proposed to represent the water and $CO_2$ leaf gas fluxes. Normally, these leaf gas exchange models are composed by a leaf photosynthesis model that calculates the $CO_2$ assimilation rate and a stomatal conductance model. The most widespread leaf photosynthesis model is the one developed by Farquhar, von Caemmerer and Berry (FvCB) (Farquhar et al., 1980). This model is generally coupled with a stomatal conductance model



such as the one proposed by Ball et al. (1987) and Collatz et al. (1991). Another leaf photosynthesis model commonly used is
the one proposed by Goudriaan et al. (1985) (G85 model). This model is used as part of the photosynthesis-stomatal description
called A-$g_s$ model developed by Jacobs (1994). The latter can be found in the LSMs of several atmospheric models such as the
European Centre of Medium-Range Weather Forecasts (ECMWF) (Boussetta et al., 2013), the Earth system model operated by
the Centre National de Recherches Météorologiques (CNRM-ESM1) (Calvet et al., 1998; Masson et al., 2013; Séférian et al.,
2016), the Dutch Atmospheric Large-Eddy Simulation (Vilà-Guerau de Arellano et al., 2014) and the CLASS mixed-layer
model (Vilà-Guerau de Arellano et al., 2015). A recent intercomparison between FvCB and G85 model (van Diepen et al.,
2022) revealed that despite fundamental differences in model structures, they have remarkable functional similarities.

The canopy level, as we have defined it, is composed by all the leaves and other phytomass that composes the plant canopy,
the soil and the air inside and right above the canopy. To connect the leaf level fluxes to the plant canopy level fluxes, assump-
tions about the vertical variability of (1) in-canopy environmental variables, (2) leaf physiology and (3) phytomass allocation
have to be made. A traditional way of representing the plant canopy is the so called "big leaf approach" in which the plant
canopy is assumed to be a homogeneous, single layer of vegetation with no vertical structure. In that approach, to calculate a
bulk surface stomatal conductance that represents the single "one big leaf" layer of the canopy and enables the calculation of
ET and NEE, the leaf gas exchange model is integrated over leaf area assuming that radiation is the only in-canopy variable
that varies vertically, and which is generally assumed to decay exponentially with leaf area index (e.g., Ronda et al. 2001).

The atmospheric boundary layer (ABL) is typically defined as "that part of the troposphere that is directly influenced by the
presence of the earth's surface, and responds to surface forcings with a time scale of about an hour or less" (Stull, 1988). In
addition, the ABL also imposes forcings on the surface. For example, the presence of clouds alters the radiation received at the
surface (Mol et al., 2023), which in turn, alters the surface fluxes of heat, water and $CO_2$. Other ABL processes such as ad-
vection of air masses with different thermodynamic properties and the entrainment of dry air masses from the free troposphere
have been reported to alter the surface turbulent fluxes (Tolk et al., 2006; Mangan et al., 2023; van Heerwaarden et al., 2009).

The objective of this research is to provide a framework with a new proposed analytical method to analyze the diurnal
dynamics of the fluxes of water and $CO_2$. In particular, we aim to answer the following research question:

- To what extent do the diurnal dynamics of environmental variables affect the diurnal dynamics of the water and $CO_2$
  exchange at leaf and canopy level?

Because processes interact across our three defined levels (leaf, canopy and ABL levels) and together they shape the ex-
change of water and $CO_2$, the framework combines observations and an encompassing model with representations at leaf,
canopy and ABL levels. The analytical method consists in the calculation of the tendency equations of the stomatal conduc-
tance, the leaf net $CO_2$ assimilation and the leaf transpiration. The tendency equations quantify the influence of temporal
changes of four environmental variables on temporal changes of the leaf gas exchange. These four environmental variables are
the photosynthetic active radiation (PAR), the air temperature (T), the vapor pressure deficit (VPD) and the atmospheric $CO_2$
concentration ($C_a$). The quantification of their influence on the leaf gas exchange allow us to investigate which environmental
variables control the dynamics of the exchange under different environmental conditions and moments of the day. This frame-



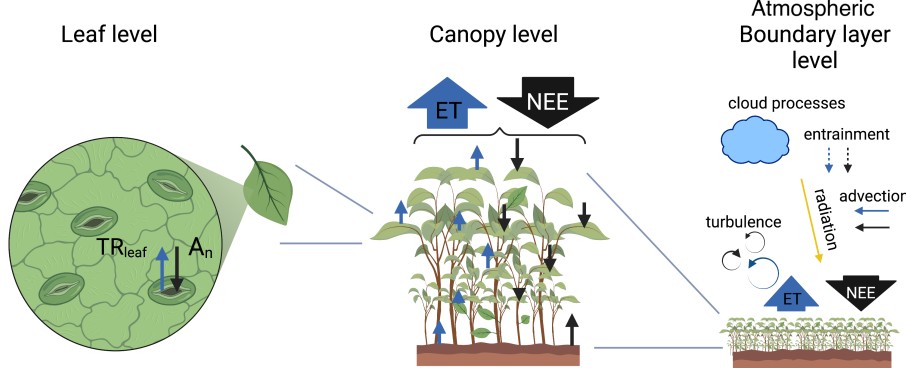

**Figure 1.** Scheme of the three levels considered to study the exchange of water (represented in blue arrows) and carbon (represented in black arrows): (1) leaf level, (2) canopy level and (3) atmospheric boundary layer. Solid lines represent a positive contribution whereas dashed lines represent a negative contribution. We have represented advection of moist and $CO_2$ enriched air. Note that the opposite can also occur. The entrainment of free tropospheric air generally introduces drier and $CO_2$ depleted air from the free troposphere and that is why it is represented with dashed lines.

work was applied to an alfalfa crop in Spain during one day in summer of 2021. To further explore the dependencies of the water and $CO_2$ exchange on the environmental variables, three modelling experiments with ABL perturbations were analyzed.

## 2 Methods

### 2.1 LIAISE field campaign and definition of the case study

This research builds on the observations obtained as part of the field campaign LIAISE (Land surface Interactions with the Atmosphere over the Iberian Semi-arid Environment). LIAISE took place in the Ebro River basin in northeastern Spain with an special observation period (SOP) from $15^{th}$ to $29^{th}$ of July 2021 with the overarching objective "to improve the understanding of land-atmosphere-hydrology interactions in a semi-arid region characterized by strong surface heterogeneity owing to contrasts between the natural landscape and intensive agriculture" (Boone et al., 2021). Further details about the field campaign and study site can be found in Boone et al. (2019), Boone et al. (2021) and Mangan et al. (2023).

La Cendrosa, our main study site, was one of the seven instrumented sites within the LIAISE domain. It was even considered a "supersite" since, in addition to standard energy balance and basic meteorology measurements, it also consisted of boundary layer measurements in the form of hourly launched radiosondes, ground based remote sensing equipment and a blimp with turbulence measurements. The crop grown at La Cendrosa is alfalfa, which was regularly irrigated by gravity driven flood irrigation during LIAISE (in the nights of the $10^{th}$ and $23^{rd}$ of July 2021).

Our research focused on dynamics observed during the SOP on the $17^{th}$ July 2021. In this study, we express time series in Coordinated Universal Time (UTC). In the region, local time (LT) is UTC + 2. We chose to use UTC because, in this region,



UTC aligns better with solar time, with 12 UTC roughly corresponding to solar noon. During the studied day, there were no clouds present, there was a light wind ($\approx 2$ m s$^{-1}$ at 2 m height) and a maximum temperature of 32 ºC. The synoptical condition was characterized by a thermal low pressure system advecting warm and dry air to the study domain in the morning and early afternoon by light, westerly winds. In the late afternoon (after 18:30 UTC) a sea breeze front, la Marinada, came in from the east bringing cooler and moister air.

## 2.2 ABL level

### 2.2.1 CLASS model

CLASS mixed-layer model (Vilà-Guerau de Arellano et al., 2015) was used to create the modelling numerical experiments. CLASS model describes the ABL but it also contains a land surface scheme (Sect. 2.3.1) and a leaf gas exchange model (Sect. 2.4.1). A numerical experiment, called Control experiment, was created to represent the studied day and it was compared
with observations. Details about the model initialization values of Control experiment and the land-surface parameters are given in the following sections. Three additional numerical experiments based on Control experiment but with imposed ABL perturbations were created and their differences with Control experiment are explained in Sect. 2.6.

Regarding how CLASS represents the ABL, CLASS is based on mixed-layer theory (summarized in Sect. 2.2 of Vilà-Guerau de Arellano et al. 2015), which states that atmospheric scalar variables are well mixed in height during the daytime,
creating an atmospheric convective boundary layer (CBL). As a consequence, the CBL can be described as a bulk layer, in which scalars (such as potential temperature, specific humidity and $CO_2$ concentration) are nearly constant with height and can be described by a single mixed-layer value. The interface between the CBL and the free troposphere is described as a sharp discontinuity in the scalars (e.g., potential temperature, humidity) normally called scalar jump. The free troposphere is described as a layer in which scalars change linearly with height. For instance, potential temperature increases at a constant
lapse rate which is initially fixed in the experiment settings. CLASS also represents the surface layer (assumed to be the lower 10 % of the CBL) with Monin-Obukhov similarity theory (Monin and Obukhov, 1954) which allows for estimations of wind, temperature and specific humidity at different heights in the surface layer. Because with CLASS we describe a CBL, our analysis during the studied day was restricted from 5 UTC, when the ABL started to be a CBL, until 15:40 UTC, when the CBL started being less driven by convection and became more stable.

### 2.2.2 Data and model initialization

To characterize the ABL, vertical profiles of temperature, relative humidity and wind obtained from radiosondes were used together with wind sensors located in a tower of 50 m (at 10 m, 20 m and 45 m). The radiosondes were hourly released from 4:00 to 17:00 UTC from the site. Radiosondes were used to provide the initial conditions for the numerical experiments (see Table 1 for exact values). Estimations of boundary layer height were obtained from the radiosondes profiles using the parcel method
(Kaimal and Finnigan, 1994). Mixed-layer values for potential temperature and specific humidity were obtained averaging the profiles below the boundary layer height minus a constant entrainment zone of 50 m and excluding the surface layer assumed to



**Table 1.** General settings, radiation parameters and initial conditions of CBL variables used in the Control experiment.

| Variable | Symbol | Value |
|---|---|---|
| *Time variables* | | |
| Time step | $\Delta t$ | 5 s |
| Initial time | $t_0$ | 5 UTC (7 LT) |
| Simulation duration | $t_f - t_0$ | 16 h |
| *Geographic coordinates for radiation* | | |
| Latitude | $\phi_{rad}$ | 41.69° N |
| Longitude | $\lambda_{rad}$ | 0.96° E |
| Day of the year | DOY | 198 |
| *Convective boundary layer* | | |
| Initial boundary layer height | $h_0$ | 150 m |
| Surface pressure | $P_s$ | 101300 Pa |
| Initial potential temperature | $\theta$ | 293 K |
| Initial potential temperature jump | $\Delta\theta$ | 1.5 K |
| Initial potential temperature lapse rate | $\gamma_\theta$ | 0.012 K m$^{-1}$ |
| Initial specific humidity | q | 9.5 g$_{water}$ kg$_{air}^{-1}$ |
| Initial specific humidity jump | $\Delta q$ | -2 g$_{water}$ kg$_{air}^{-1}$ |
| Initial specific humidity lapse rate | $\gamma_q$ | 0.011 g$_{water}$ kg$_{air}^{-1}$ |
| Initial CO$_2$ | CO$_2$ | 470 ppm |
| Initial CO$_2$ jump | $\Delta$CO$_2$ | -65 ppm |
| Initial CO$_2$ lapse rate | $\gamma_{CO_2}$ | 0 ppm m$^{-1}$ |

be the lower 10 % of the boundary layer. Lapse rates were obtained by calculating the slope of a linear regression of the vertical profile from 200 m above the top of the ABL until 3000 m height. Then, the jumps were calculated as the difference between the interpolated value at the boundary layer height and the mixed-layer value. Because the radiosondes did not measure CO$_2$,

we initialize the mixed layer value with the value provided by the CO$_2$ sensor that was part of an eddy-covariance (EC) system located at 3 m height. The initial jump of CO$_2$ and the CO$_2$ lapse rate were chosen in order to reproduce the magnitude of the diurnal variability of atmospheric CO$_2$ measured by the EC system. Advection of heat and moisture was included based on estimations derived from a network of meteorological stations operated by the Servei Meteorològic de Catalunya as calculated in previous research developed by Mangan et al. (2023). The advective terms of Control numerical experiment can be found

in Appendix A. Initial conditions for the CBL together with the time variables and the geographic coordinates needed for the initialization of radiation are given in Table 1.



## 2.3 Canopy level

### 2.3.1 Land-surface scheme

CLASS represents the plant canopy as a homogeneous single layer of phytomass without vertical structure (often referred
to as "one big leaf" approach). That layer is represented by a bulk surface canopy conductance to water vapor ($g_{surf}$) or by its
inverse, the surface canopy resistance to water vapor ($r_{surf} = \frac{1}{g_{surf}}$). To obtain $g_{surf}$, the stomatal conductance to water vapor
($g_s$) at leaf level calculated with the leaf gas exchange model of CLASS (the A-$g_s$ model, Sect. 2.4.1) is up-scaled from the
leaf to the canopy level. This upscaling is carried out by integrating $g_s$ over the leaf area and assuming an exponential decay
of PAR with respect to leaf area index as it was developed in Ronda et al. (2001). Then, to calculate ET the Penman-Monteith
equation is used:

$$ET = \frac{1}{L_v} \frac{(R_n - G)\frac{dq_{sat}}{dT} + \frac{\rho c_p}{r_a}(q_{sat} - <q>)}{\frac{dq_{sat}}{dT} + \frac{c_p}{L_v}\left(1 + \frac{r_{surf}}{r_a}\right)} \tag{1}$$

where $R_n$ is the net radiation, $G$ is the soil heat flux, $q_{sat}$ is the saturated specific humidity, $<q>$ is the well mixed specific
humidity, $c_p$ is the heat capacity of air at constant pressure, $L_v$ is the latent heat flux of vaporization, $r_a$ is the areodynamic
resistance and $\rho$ is the air density.

NEE is calculated as the difference between the net $CO_2$ rate assimilated by the vegetation-canopy ($A_{nc}$) and the soil
respiration (Resp).

$$NEE = -A_{nc} + Resp \tag{2}$$

NEE is considered negative if $CO_2$ is removed from the atmosphere and positive if $CO_2$ is added to the atmosphere. Resp
has the same sign convention and it is always positive because it adds $CO_2$ to the atmosphere. Unlike NEE and Resp, $A_{nc}$
is defined positive if $CO_2$ is removed from the atmosphere. $A_{nc}$ is derived with an approximation of Fick's law of diffusion,
eq. (3), that considers the difference between $C_a$ and the inter-cellular $CO_2$ concentration ($C_i$), the surface resistance and the
aerodynamic resistance. The $C_i$ value is calculated with the A-$g_s$ model (presented in Sect. 2.4.1).

$$A_{nc} = \frac{C_a - C_i}{r_a + 1.6 \cdot r_{surf}} \tag{3}$$

The factor of 1.6 accounts for the different molecular diffusivity of water vapor and $CO_2$ (Jacobs and de Bruin, 1997).

In the model, soil is represented with a two soil restore force model as developed by Noilhan and Planton (1989) with a
plant water stress function added by Combe et al. (2016). Soil respiration is parameterized as a function of soil temperature.
The surface and soil parameters used for the Control experiment are described in Table 2.





### 2.3.2  Data and model initialization

To characterize the canopy, we measured the leaf area index (LAI), the canopy height and time series of environmental variables (PAR, T, specific humidity, $C_a$ and wind), of soil respiration and of the turbulent surface fluxes of water and $CO_2$. We measured LAI with a LAI ceptometer (ACCUPAR LP-80). The instrument contains a PAR sensor to be deployed above the canopy together with a linear array of PAR sensors to be deployed inside the canopy. It calculates LAI considering the sun position and an spherical leaf angle distribution. We measured LAI between 10 and 12 times at different orientations in 12 locations at la Cendrosa site. In total, we obtained 132 measurements for the studied day with an average value of 1.33 and a 185 standard deviation of 0.58. Similarly, we determined the canopy height by measuring 20 times at random locations in the field. The average canopy height was 28.5 cm with an standard deviation of 6.3 cm. We measured soil respiration with a SRC-2 Soil Respiration Chamber connected to a EGM-5 Portable $CO_2$ Gas Analyzer. We measured soil respiration at 7 times during the day ranging from 7:15 to 19:00 UTC at locations close to the EC tower. Every time, three to four soil respiration measurements were recorded. As a result, we obtain seven averaged values with its corresponding standard deviation.

Time series of environmental variables and fluxes were measured. PAR and $C_a$ were measured above the canopy (at approximately 3 m) whereas temperature and specific humidity were also measured at different heights inside and right above the canopy. The sensible heat flux (H), the latent heat flux (LE) and net ecosystem exchange (NEE) were measured at a surface station that was composed of an EC system (at 3 m height), four-stream radiometers, which measured net radiation ($R_n$), and ground heat flux (G) measurements. The average energy budget non-closure was calculated as $(R_n - G - H - LE)/(R_n - G)$. 195 For more details about the set-up of the surface station the reader is referred to Mangan et al. (2023).

To compare model results and observations, we calculated the square of the Pearson correlation coefficient ($r_2$), the p-value and the root mean squared error (RMSE) between model and observed NEE and ET.

### 2.4  Leaf level

#### 2.4.1  A-$g_s$ model

In the numerical experiments created with CLASS model, we represented the leaf gas exchange with the A-$g_s$ model (Goudriaan et al., 1985; Jacobs, 1994). The exact details about the A-$g_s$ implementation used in CLASS can be found in Appendix A of Ronda et al. (2001) and in Appendix E of Vilà-Guerau de Arellano et al. (2015). The A-$g_s$ model calculates the internal $CO_2$ concentration, the net assimilation rate ($A_n$) and the stomatal conductance to water vapor ($g_s$) and to carbon dioxide ($g_{sc}$). Similar to $A_{nc}$, $A_n$ is defined positive if $CO_2$ is up-taken from the atmosphere. $A_n$ is calculated with the model 205 developed by Goudriaan et al. (1985) which captures dependencies with PAR, T and $C_i$ and requires some parameters that describe the photosynthetic traits of the vegetation. $C_i$ is calculated as a function of $C_a$, T and VPD. To use the A-$g_s$ scheme in CLASS, five environmental variables are needed: PAR, T, VPD, $C_a$ and the soil water content at the root zone ($w_2$). To represent the leaf level fluxes we have used the PAR received above the canopy (representing a sun-lit leaf), the soil water content of the second layer of soil (deeper layer of soil from CLASS) and $C_a$, T and VPD at 0.105 m height. Finally, we have 210 derived the leaf transpiration ($TR_{leaf}$) taking into account $g_s$ and VPD.





**Table 2.** Surface and soil parameters used in the Control experiment.

| Variable | Symbol | Value |
|---|---|---|
| *Surface properties* | | |
| Vegetation cover fraction | $c_{veg}$ | 1 |
| Leaf area index | LAI | 1.33 |
| Albedo | $\alpha$ | 0.2 |
| Surface skin temperature | $T_s$ | 293 K |
| *Soil properties* | | |
| Volumetric water content top soil layer | $w_g$ | 0.21 |
| Volumetric water content deeper soil layer | $w_2$ | 0.30 |
| Soil temperature at top soil layer | $T_{soil}$ | 293 K |
| Soil temperature at deeper soil layer | $T_2$ | 289 K |
| Thermal diffusivity of the skin layer | $\lambda$ | 50 |
| Respiration at 10 ℃ | $R_{10}$ | 2.73 $\mu$mol $CO_2$ m$^{-2}$ s$^{-1}$ |
| Activation energy | $E_0$ | 5.33·10$^4$ kJ kmol$^{-1}$ |

### 2.4.2  Data and model initialization

In terms of in-situ plant ecophysiology observations, a LI-6400XT Portable Photosynthesis System was used to quantify the photosynthetic traits of the alfalfa crop and to measure the diurnal variability in $g_s$. Photosynthetic traits were derived from photosyntesis response curves to PAR and to $C_i$, known as A-PAR and A-$C_i$ response curves. Three A-PAR curves

were measured at ambient temperatures ranging from 21 ℃ to 27 ℃ among the three curves, and constant ambient $CO_2$ concentration of 400 ppm. The PAR approximate set points were 0, 10, 20, 40, 60, 120, 250, 500, 1000, 1200 and 1500 $\mu$mol photons m$^{-2}$. Five A-$C_i$ response curves were measured at saturating light ($\approx$ 1500 $\mu$mol photons m$^{-2}$) and ambient leaf temperatures ranging from 21 ℃ to 28.5 ℃ with $C_i$ approximate set points of 50, 75, 100, 125, 175, 250, 400, 600, 800, 1000 and 1200 ppm. These measurements allowed the calculation of parameters that constrain the specific photosynthesis response

of the alfalfa crop. The three fitted parameters, which are in the A-$g_s$ scheme, were: (1) the $CO_2$ maximal primary productivity at 298 K ($A_{max,298}$), (2) the mesophyll conductance at 298 K ($g_{m,298}$) and (3) the light use efficiency at low light conditions ($\alpha_0$). Finally, another parameter called the high reference temperature to calculate mesophyll conductance was increased to better reflect the warm growth conditions of the alfalfa crop. All parameters of the A-$g_s$ model used in our study are indicated in Table 3. The observed and modelled response curves can be found in Appendix A.

The second type of measurements were diurnal time series of $g_s$. We measured 221 leaves from 5:30 UTC until 20:00 UTC. To mimic the field conditions we set PAR inside the chamber to the values measured outside. In practise, PAR values were updated in varying time steps of 15 minutes to one hour to reflect the values measured by a PAR sensor located above the canopy. Based on the $g_s$ and on in-canopy sensors present on the field, diurnal curves of $TR_{leaf}$ and $A_n$ were derived. These





measurements have been termed as post-processed observations. The processing procedure is based on Fick's law of diffussion
applied to the stomatal pores, assuming the thermal equilibrium between the air temperature measured inside the canopy and
the leaves and a neglegible leaf boundary layer resistance. Further details about the procedure to calculate the post-processing
of observations can be found in Appendix A.

Similarly to the canopy level, we calculated $r^2$, the p-value and RMSE to compare model results and observations. Additionally, to facilitate the visual comparison of the time series, a simple moving average was computed by calculating an unweighted
average considering the 15 previous and the 15 posterior observations for each leaf gas exchange data-point.

**Table 3.** Parameters of the A-$g_s$ model that were used for the numerical experiments. Parameters shown in bold font are the ones that were
modified from the default values used in CLASS model (Vilà-Guerau de Arellano et al., 2015). For those modified parameters, the default
values are shown in bracket and in normal font.

| Variables | | Values |
|---|---|---|
| Symbol | Definition | |
| $a_d$ (kPa$^{-1}$) | Regression coefficient to calculate $C_{frac}$ | 0.07 |
| $f_0$ (-) | Maximum value of $C_{frac}$ | 0.89 |
| $g_{min}$ (-) | Cuticular (minimum) conductance to water vapor | $0.25 \cdot 10^{-3}$ |
| **$A_{max,298}$ ($\mu$mol CO$_2$ m$^2_{leaf}$ s$^{-1}$)** | **CO$_2$ maximal primary productivity at 298 K** | **68.74** (50) |
| **$g_{m,298}$ (mm s$^{-1}$)** | **Mesophyll conductance at 298 K** | **10.2** (7.0) |
| **$\alpha_0$ (mg J$^{-1}$)** | **Light use efficiency at low light conditions** | **0.0265** (0.0170) |
| $\Gamma_{298}$ (mg CO$_2$ m$^{-2}$) | CO$_2$ compensation concentration at 298 K | 68.5 $\rho_{air}$ |
| $K_x$ (m$_{ground}$ m$^{-1}_{leaf}$) | Extinction coefficient of PAR inside the canopy | 0.7 |
| $Q_{10,CO2}$ (-) | Temperature response coefficient to calculate $\Gamma$ | 1.5 |
| $Q_{10,gm}$ (-) | Temperature response coefficient to calculate $g_m$ | 2.0 |
| $Q_{10,Ammax}$ (-) | Temperature response coefficient to calculate $A_{mmax}$ | 2.0 |
| $T_{1,gm}$ (K) | Low reference temperature to calculate $g_m$ | 278 |
| $T_{1,Ammax}$ (K) | Low reference temperature to calculate $A_{mmax}$ | 281 |
| **$T_{2,gm}$ (K)** | **High reference temperature to calculate $g_m$** | **306** (301) |
| $T_{2,Ammax}$ (K) | High reference temperature to calculate $A_{mmax}$ | 311 |

## 2.5 Tendency equations for the leaf gas exchange

We derived tendency equations for the leaf gas exchange as a method to analyze the temporal dynamics of the water and CO$_2$
exchange. These equations describe the time evolution of the leaf gas exchange variables as a function of the time evolution of
the environmental variables. Three tendency equations are proposed: (1) one for $g_s$, (2) one for $A_n$ and (3) one for TR$_{leaf}$. As a
starting point to derive the tendency equations we have used the A-$g_s$ model (see Supplementary material for a full derivation).
Therefore, the set of environmental variables used in the tendency equation is the same used in the A-$g_s$ model which is:





(1) PAR, (2) $C_a$, (3) VPD, (4) air T and (5) $w_2$. Because, according to our formulation, these five environmental variables control the leaf gas exchange, we refer to them as environmental drivers (of the leaf gas exchange). Although the tendencies were calculated considering all five environmental drivers, in this research we ignore the diurnal dynamics of $w_2$ as $w_2$ can be assumed constant in time in the root zone during the case study for La Cendrosa Alfalfa field. This assumption was based on measurements of soil volumetric water content at 30 cm deep on the field which showed a diurnal variation lower than 0.01 m³ m⁻³ and on the knowledge that the roots were likely to be deeper than 30 cm. The tendency equation for a leaf gas exchange variable $Y$ (i.e., $g_s$, $A_{n,l}$ or $TR_{leaf}$) has the following form:

$$\frac{dY}{dt} = \overbrace{\frac{\partial Y}{\partial PAR}\frac{dPAR}{dt}}^{\text{Radiative term}} + \overbrace{\left(\frac{\partial Y}{\partial T}\right)_{VPD}\frac{dT}{dt}}^{\text{Temperature term}} + \overbrace{\left(\frac{\partial Y}{\partial VPD}\right)_{T}\frac{dVPD}{dt}}^{\text{Vapor pressure deficit term}} + \overbrace{\frac{\partial Y}{\partial C_a}\frac{dC_a}{dt}}^{\text{Ambient CO}_2\text{ term}} \tag{4}$$

The left hand side (LHS) of equation (4) describes the total rate of change in time of the generic leaf gas exchange variable $Y$. Taking as an example $g_s$, this first term would indicate the rate of opening or closure of the stomatal pores. The right hand side (RHS) of the equation is composed by four terms which quantify the rate of change of Y due to temporal changes of PAR, T, VPD and $C_a$. Using the same example of $Y = g_s$, the terms would quantify the contribution to the total rate of opening or closure of the stomatal pores that is attributed to the temporal changes of PAR, T, VPD or $C_a$. For instance, the radiative term of $g_s$ tendency equation in the morning could indicate how much of the stomatal opening is happening because radiation is increasing. Each of the terms on the RHS of the equation is the product of the partial derivative of Y with respect to a particular environmental driver (which give information of the sensitivity of Y to a change in the environmental driver) multiplied by the total time derivative of the environmental driver. The sub-index notation for the T term (with VPD as sub-index) and for the VPD term (with T as sub-index) indicates that the partial derivative were calculated considering constant the variable that appears as the sub-index. This notation was inspired in the thermodynamics notation for partial derivatives and although it may seem redundant because of the definition of partial derivatives, it is deemed necessary to indicate that we considered T and VPD as independent variables. Other choice of independent variables was also possible. For instance, specific humidity or water vapor pressure could have been used in replacement of VPD. This is further explored in the Supplementary material. As it is defined now, the temperature term only includes the plant physiological processes dependent on T such as the T dependency of mesophyll conductance and of maximal primary productivity. The total time derivatives of the environmental drivers (e.g., $\frac{dPAR}{dt}$) were numerically derived from the modelling experiments output.

## 2.6 Three numerical experiments with ABL perturbations

Apart from the Control experiment, three additional numerical experiments were performed. They were created to analyze how ABL processes that perturb the environmental drivers (i.e., PAR, T, VPD and $C_a$) change the diurnal dynamics of water and $CO_2$ exchange at leaf and canopy level. The new experiments were based on ABL processes that occurred at another time during LIAISE. The three experiments are: (1) PAR-CLD which represents surface radiative changes due to a cloud passage (Mol et al., 2023), (2) VPD-ENT which represents entrainment of dry air masses from the free troposphere to the ABL (van



**Table 4.** Numerical experiments of this research. The three last rows depict the three experiments with ABL perturbations.

| Cases | Perturbation | Timing [UTC] |
|-------|-------------|--------------|
| Control | None | |
| PAR-CLD | Decrease in PAR | 10:00-14:00 |
| VPD-ENT | Entrainment of drier air from the free troposphere | 6:20-15:35 |
| TEM-ADV | Advection of cold air | 11:15-15:35 |

Heerwaarden et al., 2009) and (3) TEM-ADV which represents advection of cold air masses (Mangan et al., 2023). Table 4 summarizes the type of perturbation of each experiment and the time when the perturbation was effective.

275     In the PAR-CLD experiment, surface PAR was reduced about 25 % of its value at midday representing the radiative effect of a cloud casting a shadow in the surface (Fig. 2a). The magnitude of the reduction of PAR was based on observations during another day of the campaign ($24^{th}$ July 2021) in which high clouds were present but we did not represent the fast radiative variability also present in observations. The reason of this choice is that in the used implementation of A-$g_s$, the stomata and photosynthesis rate adapt instantaneously to changes in the environment. Because of that, sudden changes in

280 radiation would create unrealistic sudden changes in leaf gas exchange variables and cause the tendency terms to diverge. In the VPD-ENT experiment, we imposed a drier free troposphere (compared to Control) which caused entrainment of drier air masses in the ABL (Fig. 2b). The magnitude of the mixing ratio in the free troposphere (2 $g_{water}$ $kg_{air}^{-1}$ drier than the Control case) represented conditions measured on other days during the campaign. Finally, in the TEM-ADV experiment, we imposed strong cold air advection (Fig. 2c). The magnitude of the cold air advection ($\approx$ -2.7 K h$^{-1}$ at its maximum) was based on the

285 estimations of cold advection associated with the sea breeze that arrived in the region at 18:30 UTC during the studied day (Mangan et al., 2023).

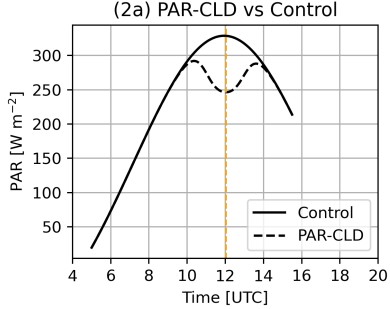
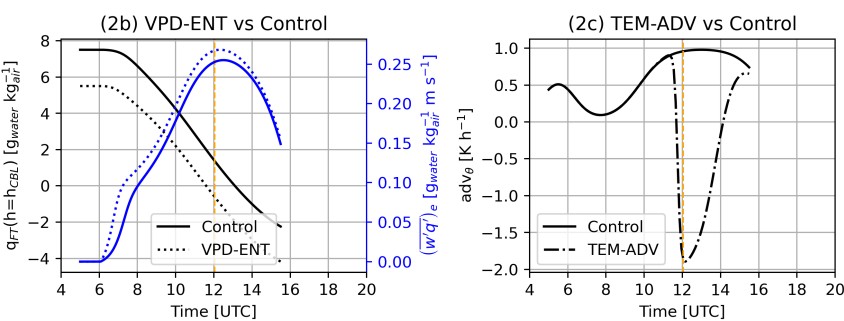

**Figure 2.** Atmospheric changes imposed in the three perturbed experiments compared to the Control simulation. Time series of: (2a) PAR for PAR-CLD and Control experiments, (2b) specific humidity of tropospheric air and entrainment flux of specific humidity for VPD-ENT and Control experiments, and (2c) temperature advection for TEM-ADV and Control experiments. Vertical dashed orange lines indicate solar noon.



To analyze the diurnal dynamics for the three perturbed numerical experiments, we compared the perturbed experiments against the Control experiment in two ways: the first was to calculate the changes in the time-averaged leaf and canopy variables and the second was to compare the changes in the tendencies terms.

The changes in the time-averaged variables were calculated as the mean percentual change of the leaf and canopy variables considering all the simulated hours. The mean percentual change, $P_{X,exp_i}$, represents the change of variable X of the i-th experiment ($exp_i$) compared to Control experiment. It was calculated as follows:

$$P_{X,exp_i} = \frac{\int_{t_{ini}}^{t_{fin}} X_{exp_i} dt}{\int_{t_{ini}}^{t_{fin}} X_{Control} dt} \cdot 100 = \frac{<X_{exp_i}>_t}{<X_{Control}>_t} \cdot 100 \qquad (5)$$

where $<X>_t$ stands for the temporal mean of variable $X$, $t_{ini}$ for the initial time and $t_{fin}$ for the final time. $P_{X,exp_i}$ was calculated for seven variables. The first three represented the leaf level and they were: $g_s$, $A_n$ and $\mathrm{TR_{leaf}}$. The second three were analogous variables but considering the whole canopy. They were: the $g_{surf}$, $A_{nc}$ and ET. Lastly, we included $-NEE$ to quantify the effects of the soil on the total carbon canopy flux. The negative sign applied to NEE was used to have the same sign convention as $A_{nc}$. We did not considered the canopy transpiration ($\mathrm{TR_{canopy}}$) because soil evaporation was negligible in our numerical experiments. As a consequence, ET was virtually equal to $\mathrm{TR_{canopy}}$.

## 3 Results

### 3.1 Control case

#### 3.1.1 Environmental drivers of the water and CO$_2$ gas exchange

During the Control day, there were no clouds and PAR was symmetric around solar noon (12:02:27 UTC) with a maximum value of 500 W m$^{-2}$ ($\approx$ 2200 $\mu$mol photons m$^{-2}$ s$^{-1}$), Fig. 3a. Observed C$_a$ was 480 ppm at 4 UTC and decreased rapidly in the morning until 8 UTC when C$_a$ stabilized around a relatively constant value of approximately 390 ppm (Fig. 3b). Observed potential temperatures varied among the heights inside (0.105 m) and above the canopy (3 m and mixed-layer value), Fig. 3c. Inside the canopy, the maximum potential temperature was acquired sooner than above the canopy. VPD was found to increase with height, acquiring a maximum difference of approximately 1000 Pa between 0.105 m and 3 m (Fig. 3d). The model captured the diurnal and height dependent variability observed for the environmental drivers excepting for C$_a$. For C$_a$ the model captured the magnitude of the diurnal variability but failed to capture the dynamics of it. Modelled C$_a$ decreased at a lower pace than observed C$_a$. As a consequence, modelled C$_a$ was larger than observed C$_a$ particularly in the morning. Possible explanations of this mismatch are discussed in Sect. 4.

#### 3.1.2 Leaf gas exchange

Observed $g_s$ showed highest values in the morning reaching a maximum value of approximately 0.020-0.030 m s$^{-1}$ ($\approx$ 1.00-1.20 mol$_{air}$ m$_{leaf}^{-2}$ s$^{-1}$) at 10 UTC and declined afterwards until the end of the day (Fig. 4a). Modelled $g_s$ was maximum



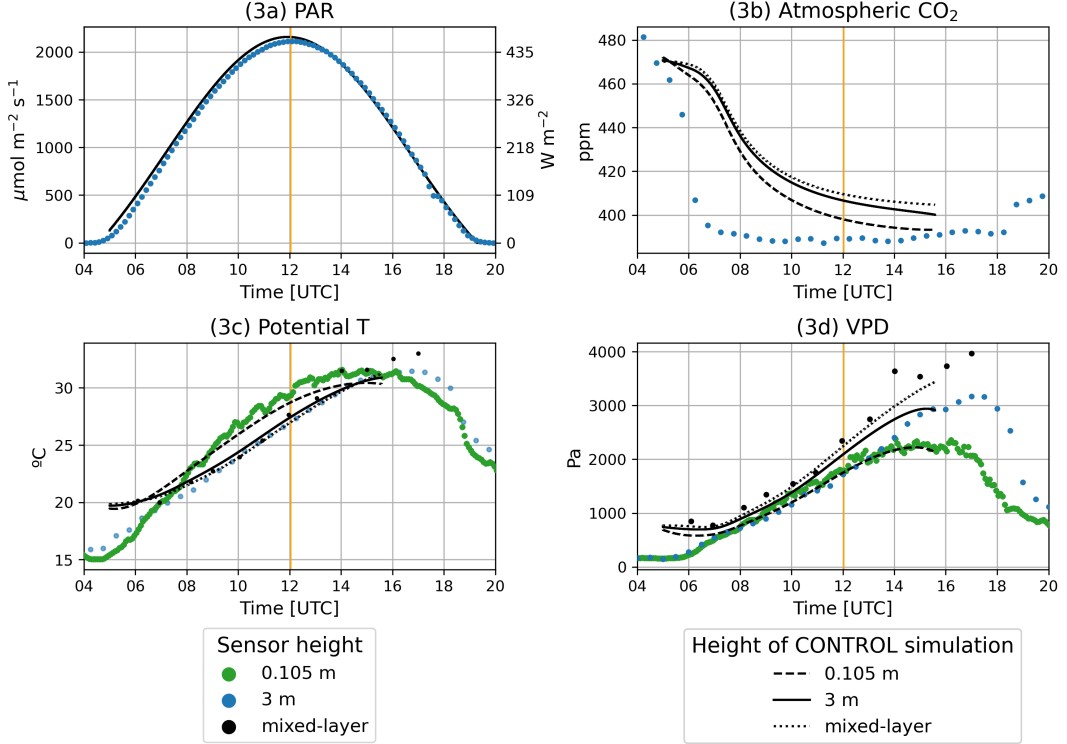

**Figure 3.** Diurnal time series of: (3a) PAR, (3b) $C_a$, (3c) potential temperature and (3d) VPD. Observations are depicted as dots. The sensor at 0.105 m height is located inside the canopy. Black dots shown potential T and VPD derived from radiosondes. The solid black lines correspond to the Control experiment at 3 m, the dashed line at 0.105 m and the dotted line is the mixed-layer value. Vertical orange lines depict the solar noon.

at the same time as the observed $g_s$ and showed relatively weak but significant correlation with the observations ($r^2$ = 0.223, p < 0.001). Observed and modelled $A_n$ followed closely the diurnal pattern of PAR (Fig. 3a), achieving maximum values approximately at solar noon between 12 and 13 UTC (Fig. 4b). Model results showed significant correlation with the post-processed observations ($r^2$ = 0.383, p < 0.001). Finally, $TR_{leaf}$ also increased in the morning and decreased in the afternoon with a maximum achieved towards the afternoon between 12:45 and 14 UTC (Fig. 4c). Model results showed significant and high correlation ($r^2$ = 0.677, p < 0.001) with post-processed observations although model results overestimated maximum $TR_{leaf}$ compared to the post-processed observations. Comparing the modelled leaf gas exchange variables with the observed moving averaged variables, we noticed that the diurnal pattern is well captured. Therefore, suggesting that relatively low correlations were partly due to the scatter of observations.





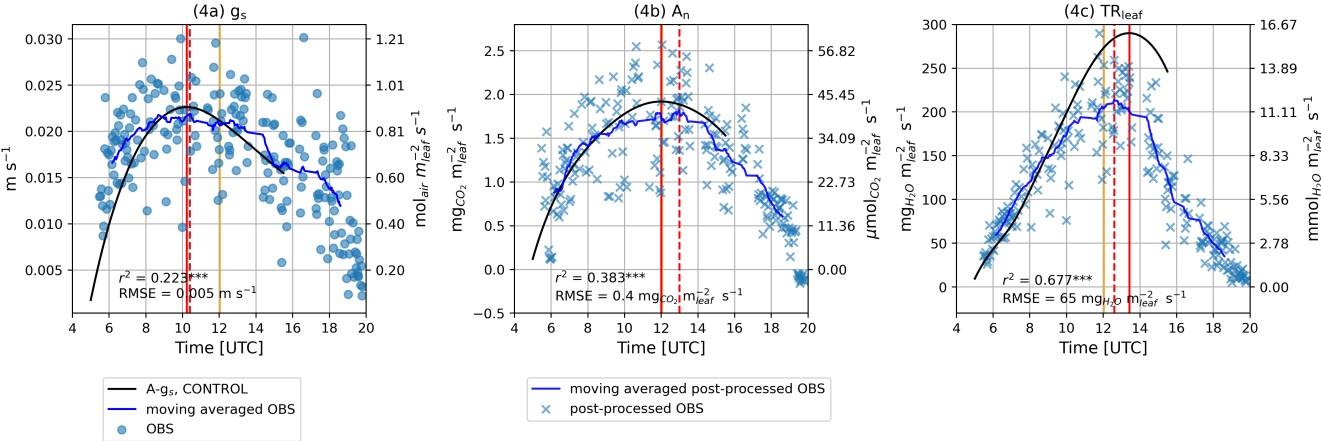

**Figure 4.** Diurnal time series of: (4a) $g_s$, (4b) $A_n$ and (4c) $TR_{leaf}$ for Control experiment. Direct observations are depicted in blue dots whereas post-processed observations are depicted in blue crosses. Moving averaged observations and post-processed observations are indicated with a blue line. Model results of the Control experiment are depicted by black solid lines. Vertical solid orange lines depict solar noon. The time when the maximum value of the leaf gas exchange variable at hand was achieved is depicted by red solid vertical lines for model results and red dashed vertical lines for observations or post-processed observations.

### 3.1.3 Canopy level gas exchange

Observed NEE was positive before 6 UTC indicating a net transport of $CO_2$ from the surface to the atmosphere (Fig. 5a) which suggests that ecosystem respiration was providing $CO_2$ to the atmosphere. From 6 until 18 UTC, the observed flux was negative acquiring a minimum value between 11 and 12 UTC. The diurnal negative NEE indicated a net transport of $CO_2$ from the atmosphere to the surface suggesting that the photosynthesis of the crop was dominant over respiration processes. Modelled NEE obtained a strong and significant correlation with observations ($r^2 = 0.89$ and p-value < 0.001). Additionally, the time at which the minimum NEE was attained matched well between observations and model results. Observed soil respiration was between 4.5 and 9 $\mu mol_{CO_2}$ m$^{-2}$ s$^{-1}$ during the day which coincided with the range reproduced by the model. Observed ET acquired a maximum value of 6.5 mmol$_{H_2O}$ m$^{-2}$ s$^{-1}$ that stayed relatively constant between 11 and 15 UTC. This plateau was not reproduced by the model which peaked at approximately 12:45 and then declined. Similarly to what was observed at leaf level, modelled ET was higher than observed ET. The overestimation of modelled ET was of the same magnitude than the observed energy budget non-closure. Despite the apparent differences, modelled ET obtained strong and significant correlation with observed ET ($r^2 = 0.95$ and p-value < 0.001) (Fig. 5).

### 3.1.4 Tendencies of the leaf gas exchange

The total tendencies of $g_s$, $A_n$ and $TR_{leaf}$ (LHS of eq. (4)) described the diurnal dynamics observed in the modelled leaf gas exchange variables (Sect. 3.1.2). These dynamics consisted of an increase of the leaf gas exchange variables (positive total





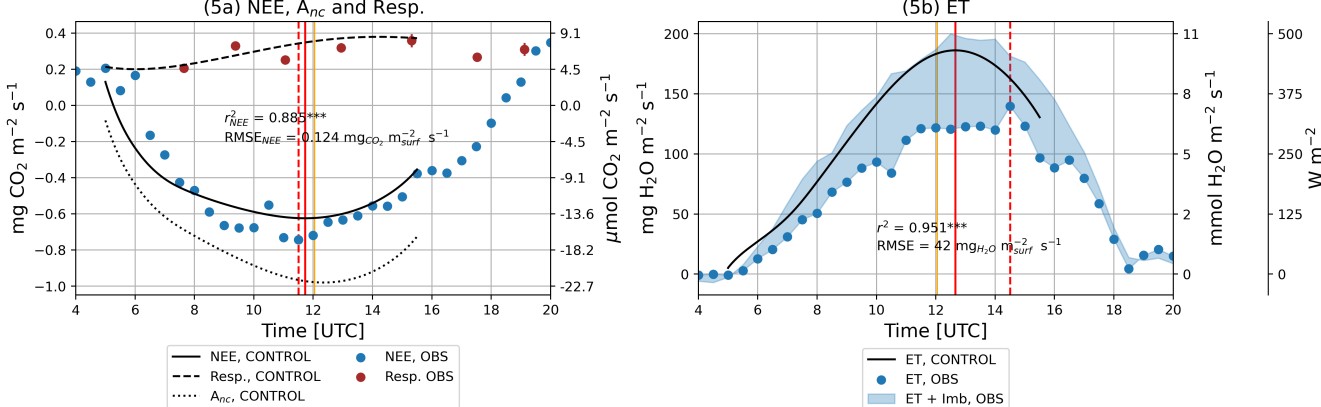

**Figure 5.** Diurnal time series of: (5a) NEE, $-A_{nc}$ and soil respiration (Resp.) and (5b) ET and latent heat flux (LE). 30 minutes averaged observations from the EC system are depicted in blue dots whereas model results are depicted with lines with different line-styles. Respiration observations are depicted in red dots with an error bar that represents the standard error of the mean. The measured surface energy budget non-closure is depicted as a shade that covers from the measured ET or LE value to the measured ET or LE value plus the measured surface energy budget non-closure. Vertical solid orange lines depict solar noon. The times when the maximum of NEE and ET was achieved are depicted by red solid vertical lines for model results and red dashed vertical lines for observations or post-processed observations.

tendency term) until reaching a maximum (null total tendency term) which occurred in the morning for $g_s$, at noon for $A_n$ and in the afternoon for $\mathrm{TR}_{leaf}$, followed by a decrease until the end of the simulation time (negative total tendency term). The sum of the partial tendency terms (RHS of eq. (4)) exactly matched the total tendency term (LHS of eq. (4)) for $g_s$, $A_n$ and $\mathrm{TR}_{leaf}$ (see the overlapping of the solid black line and the grey dashed line in Fig. 6a, 6b and 6c). This verification guaranteed that the

temporal evolution of $g_s$, $A_n$ and $\mathrm{TR}_{leaf}$ was fully determined by the temporal evolution of PAR, T, VPD and $C_a$. Focusing on the partial terms, the radiative terms (PAR terms) were found to be the primary contribution to the total terms of the leaf gas exchange variables, especially for $g_s$ and $A_n$. The temporal changes in PAR tended to increase the leaf gas exchange variables before noon (positive PAR terms) and to decrease it after noon (negative PAR terms). T, VPD and $C_a$ terms added secondary temporal dynamics to the PAR term contribution.

For $g_s$ (Fig 6a), the VPD term was negative from 7 to 15 UTC indicating that the diurnal increase in VPD (Fig. 3d) was leading to smaller $g_s$ values. This effect was partially compensated by the T and $C_a$ terms which were both positive. Therefore, both the increase of T due to the diurnal warming of the atmosphere (Fig. 3c) and the decrease in $C_a$ due to the entrainment of $CO_2$ depleted air from the free troposphere (Fig. 3b) contributed to increase $g_s$. The net effect of these opposing terms resulted in the maximum of $g_s$ being achieved 2 hours before solar noon. For $A_n$ (Fig. 6b), the diurnal increase in T favored higher $A_n$

values, increasing $A_n$ rate specially in the morning (positive T term). Both VPD and $C_a$ terms were relatively small compared to PAR and T terms which suggested that the diurnal dynamics of $A_n$ were relatively insensitive to diurnal changes in VPD and $C_a$. Finally, for $\mathrm{TR}_{leaf}$ (Fig. 6c), both T, VPD and $C_a$ terms tended to further increase $\mathrm{TR}_{leaf}$ values. T and VPD terms were found comparable and greater that the PAR term for several hours in the morning and early afternoon. The combined effect





of the T, VPD and $C_a$ terms was responsible of delaying the maximum $TR_{leaf}$ from occurring at solar noon (if the model
would be only sensitive to PAR temporal changes) to 13:30 UTC. The results of the partial tendency terms highlighted that
diurnal temporal changes in PAR primarily forced the net diurnal dynamics of the leaf gas exchange variables, whereas diurnal
temporal changes in $C_a$ were found the least important factor to describe the leaf gas exchange dynamics.

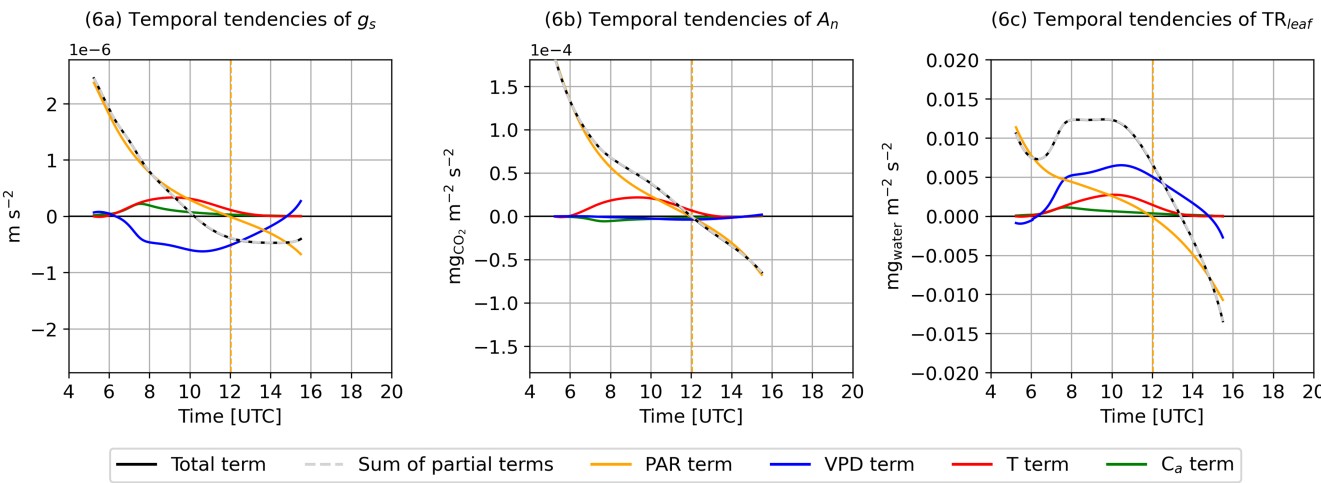

**Figure 6.** Temporal evolution of the tendencies of: (6a) $g_s$, (6b) $A_n$ and (6c) $TR_{leaf}$. Black lines depict the total tendency terms, grey dashed
lines depict the sum of the partial terms and the other solid coloured lines depict the partial tendency terms due to temporal changes of PAR
(orange lines), VPD (blue lines), T (red lines) and $C_a$ (green lines). The vertical dashed orange lines depict solar noon.

### 3.2 Three experiments with ABL perturbations

#### 3.2.1 Environmental drivers of the water and CO$_2$ gas exchange

The three experiments with ABL perturbations (PAR-CLD, VPD-ENT and TEM-ADV) modified the environmental drivers
of the water and CO$_2$ exchange compared to the Control simulation except for $C_a$, which remained almost equal to Control
for all the perturbed experiments (Fig. 7a). The PAR-CLD experiment described a cloud passage which reduced surface PAR
(Fig. 2a). The cloud passage also modified the surface energy balance during and after the cloud. As a consequence, T and
VPD were reduced up to 1 K and 250 Pa respectively (Fig. 7b and c). The VPD-ENT experiment, which described a drier
free troposphere than Control experiment, only modified VPD which increased up to 250 Pa (Fig. 7c). Lastly, the TEM-ADV
experiment, which described strong cold air advection, reduced not only T up to 4 K but VPD up to 1000 Pa (Fig. 7b and c).

#### 3.2.2 Leaf and canopy gas exchange

The changes in the environmental drivers led to changes compared to the Control experiment in the leaf and canopy variables
that describe the water and CO$_2$ exchange. The mean values of the exchange variables changed up to 11 % compared to Control



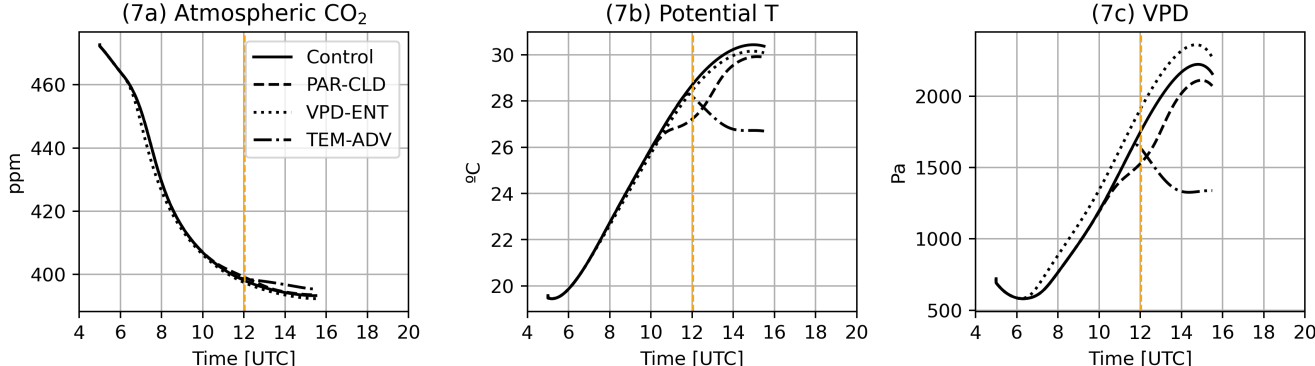

**Figure 7.** Temporal evolution of: (7a) $C_a$, (7b) potential T and (5c) VPD at 0.105 m for the four experiments. Solid black line corresponds to the Control experiment, dashed black line to PAR-CLD experiment, dotted black line to VPD-ENT experiment and dashed-dotted black line to TEM-ADV experiment. Vertical dashed orange lines depict solar noon.

experiment (Fig. 8). For PAR-CLD, there was a slight reduction of stomatal conductance and surface conductance ($P_{g_s}$, $P_{g_{surf}}$ > -3 %), a moderate reduction of the assimilated $CO_2$ by the vegetation ($P_{A_n}$, $P_{A_{nc}}$ < - 3 %) and a strong reduction in the water exchange at leaf and canopy levels ($P_{TR_{leaf}}$, $P_{ET}$ < -5 %). For VPD-ENT, a moderate reduction in stomata and surface conductance was reported ($P_{g_s}$, $P_{g_{surf}}$ ≈ -3 %) whereas $TR_{leaf}$ and ET were moderately increased ($P_{TR_{leaf}}$, $P_{ET}$ > 3 %). $A_n$ and $A_{nc}$ barely changed compared to Control experiment (|$P_{A_n}$|, |$P_{A_{nc}}$| < 2 %). Lastly, TEM-ADV reported a strong increase

in stomatal conductance and surface conductance ($P_{g_s}$, $P_{g_{surf}}$ > 7.5 %), a strong decrease in $TR_{leaf}$ ($P_{TR_{leaf}}$ < -5 %) and a moderate decrease in ET ($P_{ET}$ < -3 %). Similarly to VPD-ENT experiment, $A_n$ and $A_{nc}$ barely changed compared to Control simulation (|$P_{A_n}$|, |$P_{A_{nc}}$| < 2 %).

Comparing between experiments, all of them modified moderately or strongly $TR_{leaf}$ and ET whereas only PAR-CLD experiment modified moderately $A_n$ and $A_{nc}$. Comparing the trends between leaf and canopy level for each experiment ($g_s$

versus $g_{surf}$, $A_n$ versus $A_{nc}$ and $TR_{leaf}$ versus ET), we generally observed similar patterns in magnitude and sign of the change between leaf and canopy level variables, with two remarkable exceptions. The first exception was the small decrease in $A_n$ at leaf level opposed to a small increase in $A_{nc}$ at canopy level for VPD-ENT experiment compared to Control. Further analysis revealed that the decrease in VPD for VPD-ENT was leading to a reduced $C_i$ which affected the $CO_2$ exchange differently at the leaf and at canopy level. At leaf level, the decrease in $C_i$ implied lower maximum rates of photosynthesis

because less $CO_2$ was available to perform photosynthesis, which finally lead to a smaller net assimilation rate. However, at canopy level, the decrease in $C_i$ was accounted for with a diffusion type of equation (equation 3) which resulted in a higher $A_{nc}$ due to a higher $CO_2$ gradient ($C_a$-$C_i$). The second unexpected result was the large decrease in $TR_{leaf}$ at leaf level compared to the moderate decrease in ET at canopy level for TEM-ADV. Further exploration revealed that the change of magnitude was partially attributed to the effect of the wind in the exchange which was only accounted for at canopy level. Although horizontal

wind was equal for all experiments, the vertical component was greater for TEM-ADV than for Control which implied a smaller





aerodynamic resistance and favourable conditions for the exchange of water for TEM-ADV with respect to Control. This effect was partially responsible of a smaller decrease on ET than on $\text{TR}_{leaf}$.

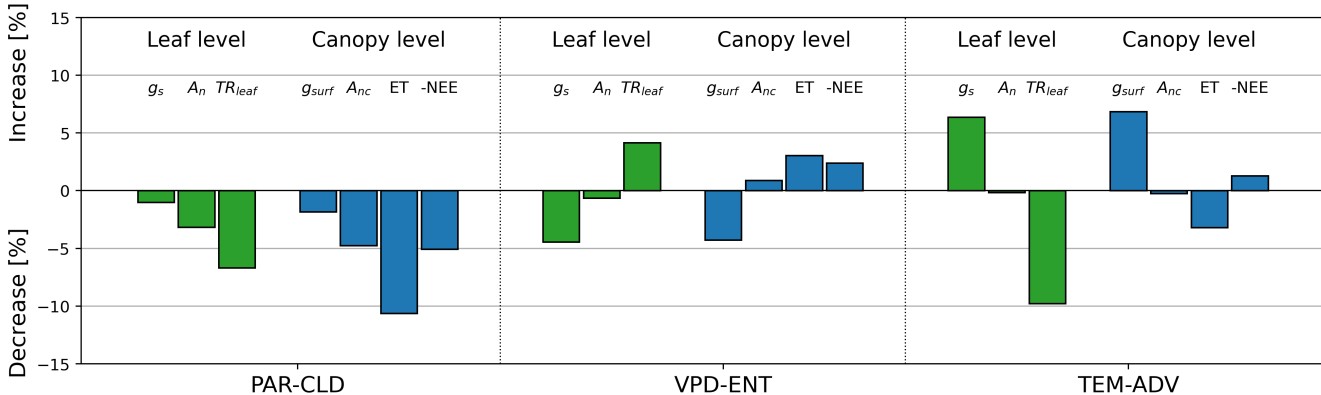

**Figure 8.** Barplot of the mean percentual change of seven leaf and canopy variables of the perturbed experiments with respect to the Control experiment. The set of variables is composed by three leaf gas exchange variables ($g_s$, $\text{A}_n$ and $\text{TR}_{leaf}$) indicated by green bars, and by four canopy gas exchange variables ($g_{surf}$, $\text{A}_{nc}$, ET and -NEE) in blue bars.

### 3.2.3 Tendencies of the leaf gas exchange

Analysing the tendency terms of the three perturbed ABL experiments showed clear separation between the effects of the environmental drivers on $\text{A}_n$ and $\text{TR}_{leaf}$ (Fig. 9). The reduction of $\text{A}_n$ for PAR-CLD experiment (Fig. 9a) followed closely the shape of the decrease in PAR which occurred roughly between 10 and 12 UTC (Fig. 2a). Accordingly, the difference of the net total tendency terms between PAR-CLD and Control followed closely the difference of the radiative terms (Fig. 9b). This suggests that the dip in $\text{A}_n$ was mostly attributed to the radiative changes. As a second order effect, the difference of the temperature terms indicated that the temperature variability during the cloud also contributed in reducing $\text{A}_n$. For VPD-ENT and TEM-ADV, $\text{A}_n$ diurnal dynamics remained quite similar to those of Control, hence the difference of the tendencies terms were much smaller that those of PAR-CLD.

Similarly to the reduction of $\text{A}_n$ for PAR-CLD, the reduction of $\text{TR}_{leaf}$ for PAR-CLD (Fig. 9e) experiment was strongly influenced by the reduction in radiation (Fig. 9f). However, in this case both the temporal dynamics of temperature and VPD during the cloud contributed to the reduction of $\text{TR}_{leaf}$ (Fig. 9f). For VPD-ENT experiment, $\text{TR}_{leaf}$ was higher than Control between 6-7 UTC and the end of the simulation. The difference in the total tendency term was very similar to the difference of the VPD term indicating that the VPD diurnal variability was responsible of the increase $\text{TR}_{leaf}$ (Fig. 9g). Lastly, $\text{TR}_{leaf}$ for TEM-ADV experiment was strongly reduced compared to Control after the advection started (Fig 9e). This reduction resulted from the contribution of both the reduction of VPD and T, with the dynamics of VPD being roughly three times more important than the dynamics of temperature (Fig 9h). Interestingly, our tendencies showed that PAR contributed positively to $\text{A}_n$ and $\text{TR}_{leaf}$ during the advection of cold air (Fig. 9d and h) even though PAR diurnal variability remained unchanged





compared the control. The positive value of the difference in PAR tendency terms for $TR_{leaf}$ indicated that temporal changes in radiation were contributing less to the decrease of $TR_{leaf}$ in TEM-ADV compared to Control. This effect was related to the lower contribution of radiative variability to the changes in stomatal conductance which were predominantly influenced by the drop of VPD.

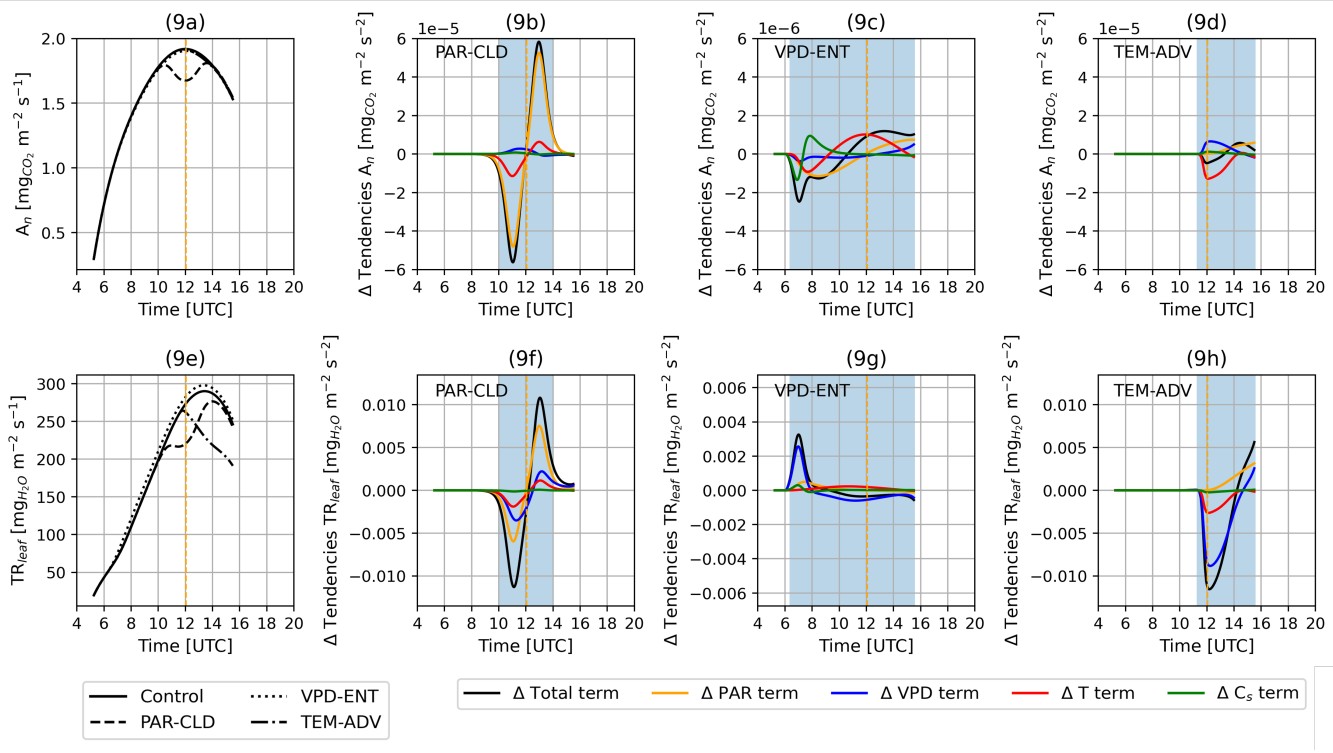

**Figure 9.** Diurnal evolution of (9a) $A_n$ and (9e) $TR_{leaf}$ for Control (solid black line), PAR-CLD (dashed black line), VPD-ENT (dotted black line) and TEM-ADV (dot-dashed black line). Diurnal evolution of the difference of $A_n$ tendencies terms of each perturbed experiment against Control experiment are shown in (9b) for PAR-CLD, (9c) for VPD-ENT and (9d) for TEM-ADV. The diurnal evolution of the difference of $TR_{leaf}$ tendencies terms between each perturbed experiment against Control experiment are shown in (9f) for PAR-CLD, (9g) for VPD-ENT and (9h) for TEM-ADV. Notice that the difference of the tendencies terms for VPD-ENT experiment are smaller than for PAR-CLD and TEM-ADV experiments. Because of that, the y-axis of Fig. 9c is 10 times smaller than the y-axis of Fig. 9b and 9c and the y-axis of Fig. 9g is half the y-axis of Fig. 9f and 9h. Solar noon is indicated with a vertical dashed orange line. Time periods when the ABL perturbations were effective are shown as blue shades in panels 9b, 9c, 9d, 9f, 9g and 9h.

## 4 Discussion

The framework proposed in this research was constituted by observations and a coupled model with descriptions at leaf, canopy and ABL levels and by tendency equations of the modelled leaf gas exchange (for net assimilation rate, stomatal con-



ductance, and leaf transpiration). Our research strategy resembled that of Vilà-Guerau de Arellano et al. (2020) and Mangan et al. (2023) in that a range of spatial scales were integrated to investigate the diurnal variability of turbulent fluxes. Additionally, a key element of the study is the comprehensiveness of the measurements (leaf gas exchange observations, surface turbulent fluxes and atmospheric boundary layer observations) which is considered suitable to progress in the investigation of the vegetation/ecosystem response to meteorological conditions and the effect of ecosystem responses on the atmospheric dynamics (land-atmosphere bi-directional feedbacks) (Helbig et al., 2021).

In this study, the coupled model CLASS could reproduce the observed diurnal variability of the environmental drivers for the studied day excepting for the variability of $C_a$ (Fig. 3b). Unlike for VPD and T, $C_a$ measurements were only available at 3 m and we did not have information about its vertical variability. CLASS model assumes that $C_a$ is well-mixed since the start of the numerical experiment. However, $C_a$ vertical profiles can depict strong vertical gradients during and after the morning transition from a stable ABL to an unstable and well-mixed ABL as it has been previously observed over grass (Casso-Torralba et al., 2008). As a consequence, the initial observed $C_a$ values may not be representative of the initial convective ABL. Nonetheless, we have performed a sensitivity analysis of the results of this research to the modelled $C_a$ (not shown) and we have found no significant impact on the conclusions.

Regarding the leaf level, observations of stomatal conductance were scattered which led to scatter in the post-processed observations of net assimilation rate and leaf transpiration (Sect. 3.1.2). This reflected that randomly picked leaves within the alfalfa canopy at a same moment of the day gave values of stomatal conductance that differ from each other. We attribute this spread to the different environmental conditions experienced by each leaf (e.g., sun-lit and shaded leaves) and to differences in leaf properties (e.g., age or damage of leaves). Similar dependencies of leaf gas exchange on sun or shade preconditioning of leaves and on the age of the leaves have been previously reported for a cotton crop by Echer and Rosolem (2015). Our modelled results did not represent this variability as they were based on a single combination of the modelled PAR, atmospheric $CO_2$, T and VPD per time during the day and the same photosynthetic traits for all leaves. Despite the scatter, the moving averaged observations presented a magnitude and diurnal characteristics that were consistent with the model results such as the time of occurrence of maximum values. Based on the modelled leaf gas exchange, tendency equations were used to quantify the effect of the diurnal dynamics of the environmental drivers on the dynamics of the leaf gas exchange. In that regard, the tendency terms informed about the modelled leaf gas exchange and are bounded by the limitations of the same. Some additions that could be included to the A-$g_s$ scheme and tendency equations, but were not in this research, are the distinction between direct and diffuse light (Gu et al. 2002; Pedruzo-Bagazgoitia et al. 2017; Durand et al. 2021) and the temporal adaptation of the stomata to instantaneous changes in environmental conditions (Sellers et al. 1996; Vico et al. 2011; Sikma et al. 2018). Sensitivities to these two additions may be important to generalize findings about cloud-vegetation interactions specially for fast radiative perturbations as those that have been observed and modelled in previous research (Kivalov and Fitzjarrald 2018 and Mol et al. 2023).

Tendency equations, similar to the ones presented here, have been proposed in the past for leaf transpiration (Jarvis and McNaughton, 1986), evapotranspiration (van Heerwaarden et al., 2010) and net ecosystem exchange (Pedruzo-Bagazgoitia et al., 2017) but with substantial differences with respect to our study. Jarvis and McNaughton (1986) used a similar approach



to investigate the dependency of transpiration on stomatal conductance for scales ranging $10^{-5}$ m up to $10^5$ m. The approach was different because it was not intended to analyze temporal dynamics of the fluxes but to investigate the sensitivity of transpiration on stomatal conductance at different scales. Because of that, Jarvis and McNaughton (1986) used differential equations but not with respect to time. Additionally, the $CO_2$ fluxes were not investigated and to the author's knowledge, this is the first time that the tendencies have been calculated simultaneously for stomatal conductance, leaf transpiration and net assimilation rate providing a complete view of the leaf gas exchange. On the other hand, van Heerwaarden et al. (2010) calculated tendency equations for the canopy evapotranspiration and Pedruzo-Bagazgoitia et al. (2017) for the gross primary productivity. Both of them applied the approach to investigate diurnal dynamics in realistic field conditions. The approach of van Heerwaarden et al. (2010) was based on Penman–Monteith equation combined with mixed-layer theory for CBL whereas Pedruzo-Bagazgoitia et al. (2017) was based on the up-scaled $CO_2$ flux given by A-$g_s$ (eq. (3)). The main difference between those approaches and the one presented here is that we calculated the terms as a function of state primary variables whereas van Heerwaarden et al. (2010) and Pedruzo-Bagazgoitia et al. (2017) did it for intermediate variables. For example, a term of the equation proposed by Pedruzo-Bagazgoitia et al. (2017) contained the temporal derivative of $C_i$ which may be more difficult to interpret and relate to environmental processes than changes in PAR, T, VPD and $C_a$. We acknowledge that for certain research questions it may be relevant to use a different subset of independent variables. However, the choice of another subset is also possible within the proposed framework.

Returning to the initial research question (*To what extent do the diurnal dynamics of environmental drivers affect the diurnal dynamics of the water and $CO_2$ exchange at leaf and canopy level?*), we observed that the dynamics of stomatal conductance and net assimilation rate were primarily forced by the diurnal dynamics of radiation. As a second order effect, the dynamics of net assimilation rate were affected by those of T and the dynamics of stomatal conductance by those of T and VPD. Leaf transpiration was affected to a similar extent by the dynamics of PAR, T and VPD, with PAR dynamics being the most important factor in the early morning and late afternoon. Leaf gas exchange dynamics were less sensitive to dynamics of atmospheric $CO_2$ concentration compared to dynamics of PAR, T and VPD. These results indicates that radiative perturbations (as those created by clouds shade) strongly affect the diurnal evolution of the assimilation rate, fact that was further explored by PAR-CLD experiment. In fact, from all experiments PAR-CLD was the one that modified most the net assimilation rate and gross primary productivity, suggesting that the representation and understanding of clouds and its effects on the surface are a crucial factor to understand and represent diurnal variability of $CO_2$ fluxes (Vilà-Guerau de Arellano et al., 2023). Additionally, PAR-CLD experiment showed that not only radiative changes but also associated temperature changes produced by a cloud shade can further reduced the net assimilation rate. Although the other experiments which represented entrainment of drier air and advection of cold air (VPD-ENT and TEM-ADV) did not significantly modified the net assimilation rate and gross primary productivity, they did modify the stomatal conductance, surface conductance, leaf transpiration and evapotranspiration. Similar to what was reported by van Heerwaarden et al. (2009), entrainment of dry air under non-stressed soil water availability (VPD-ENT experiment) enhanced the water surface exchange. Lastly, the cold air advection experiment (TEM-ADV experiment) suggested that heat advection modifies leaf transpiration not only because the temperature changes but also due to the associated VPD changes.



To close the discussion, we will mention some possible avenues for future work. We envision that the tendency equations can help to identify errors and shortcomings, and to investigate limiting factors of the water and $CO_2$ exchange, represented in
global models that explicitly include vegetation (Doutriaux-Boucher et al., 2009), in standard weather and climate land-surface models (Renner et al., 2021) and/or in new generation models such as land-surface models with multi-layer canopy (Bonan et al., 2021). One possible application of the method is to investigate the dependency of the temporal dynamics of the leaf gas exchange on soil moisture during a dry spell (Combe et al., 2016) or during and after a precipitation event. Soil moisture tendencies were not investigated in this manuscript because the alfalfa crop leaf gas exchange was not limited by its root soil
water content. However, our modelling framework enables the calculation of the soil water content tendency. In principle, the tendency equations can be applied to timescales larger than one day as weekly or monthly scale. Another application of the tendencies could be to analyze how the relations between the environmental drivers and the leaf gas exchange vary vertically inside a canopy. For instance, this analysis could help in understanding the causes of the different magnitude of the fluxes in the layers of a forest (e.g., understory versus top of the canopy). For this, the method could be applied to different layers within a
multi-layer canopy (Bonan et al. 2021; Pedruzo-Bagazgoitia et al. 2023). Lastly, we would like to comment that the tendencies can be calculated with the output of models if the needed variables have been saved. Because of that, this interpretative tool is not computationally expensive once the model has been run.

## 5 Conclusions

In this research, we investigated the leaf and canopy exchange of water and $CO_2$ and its relationship with the diurnal dy-
namics of four environmental variables which were: photosynthetic active radiation (PAR), air temperature (T), vapor pressure deficit (VPD) and atmospheric $CO_2$ concentration ($C_a$) for an irrigated alfalfa field. We based the research on one day during the field campaign LIAISE (Land Surface Interactions with the Atmosphere in the Iberian Semi-Arid Environment). We created a Control numerical experiment based on the studied day with a mixed-layer model (CLASS model; Vilà-Guerau de Arellano et al. 2015) that represents the convective atmospheric boundary layer (ABL) level, the canopy level and the leaf level (with
the A-$g_s$ model; Goudriaan et al. 1985; Jacobs 1994). In terms of observations, the leaf gas exchange was characterized with observations carried out with a LI-6400XT Portable Photosynthesis System. The canopy gas exchange was characterized with 30 minute averaged Eddy Covariance (EC) measurements and soil respiration measurements. To quantify the contributions of the diurnal dynamics of the environmental variables (PAR, T, VPD and $C_a$) to the water and $CO_2$ exchange, we derived three tendency equations for stomatal conductance, net assimilation rate and leaf transpiration. To investigate the effects if
ABL processes on the local exchange, we created three additional numerical experiments with three ABL perturbations: (1) a reduction in surface radiation due to a cloud shade, (2) entrainment of drier air masses from the free troposphere and (3) a strong cold air advection.

We ascertain that the partial tendency terms of the leaf gas exchange fully accounted for the diurnal dynamics of the leaf gas exchange. An important finding was that PAR diurnal dynamics strongly influenced the diurnal dynamics of stomatal conduc-
tance and assimilated carbon dioxide. When investigating the water and $CO_2$ exchange under the three perturbed experiments,



we found that all experiments modified to a similar extent the exchange of water whereas only the experiment of the decrease in surface radiation due to a cloud shade modified significantly the $CO_2$ exchange. The analysis with the tendency equations revealed first-order effects (e.g., radiation reduction due to a cloud shade diminishes net assimilated $CO_2$) and second-order effects (e.g., the reduction of air temperature due to the cloud shade enhances the decrease in assimilated $CO_2$ due to less surface

radiation) of the ABL perturbations on the exchange. We envision multiple applications of the proposed tendency equations, all of them orientated to support the interpretation of model results of the exchange of water and $CO_2$ between the vegetation and atmosphere, and to investigate limiting and controlling factors of the exchange.

*Data availability.*    LIAISE products are available in the following database: https://liaise.aeris-data.fr/products/. Any other data needed to replicate this research will be made available on request.



**Appendix A: Additional information about methods**

**A1 Heat and moisture advection**

As mentioned in the manuscript, advection was estimated with measurements of wind, temperature and specific humidity from an atmospheric weather station network operated by the Servei Meteorològic of Catalunya. For more details on how the advection was calculated the reader can refer to Sect. 4.2 "Mixed layer data: Model initialization & advection" and Appendix B of the manucript written by Mangan et al. (2023). The unique differences between the cited manuscript and the present manuscript are: that we used the daily advection estimation of the $17^{th}$ July 2021 instead of the monthly average and that we smoothed the advection term by fitting the estimations to a continuous function. We smoothed the estimated advection to ensure that the temporal evolution of the environmental variables would be differentiable. This was desired to facilitate the interpretation of the tendency terms. The continuous function we used was:

$$adv_Y(t) = \sum_{i=1}^{i=2} A_{Y,i} \left( \frac{1}{1 + e^{-2k_{1,Y,i}(t-t_{ini,i})}} - \frac{1}{1 + e^{-2k_{2,Y,i}(t-t_{fin,i})}} \right) \; ; \tag{A1}$$

where $Y = \theta, q$ and $i$ represents the different advection regimes. In our case the regimes were two: (1) the warm and very dry regime from 5 to 8 UTC and (2) the very warm and slightly dry regime from 9 till 18 UTC. Coefficients $A_{Y,i}$, $t_{ini,i}$, $t_{fin,i}$, $k_{1,Y,i}$ and $k_{2,Y,i}$ represent the amplitude of the advection, the initial and final time when advection occurs and the rate of change from no advection to advection and viceversa. Note that the sea breeze (advection of cold and wet air from 18 UTC onwards) was not imposed as a boundary condition in the simulations because at this time the atmosphere was not well-mixed and because of that, we did not study that period. However, the smooth function of the sea breeze advection was also calculated for the temperature (not shown here) as it was imposed for TEM-ADV experiment.

The estimations of heat and moisture advection together with the smoothed advection terms used as boundary conditions in the Control experiment are shown in Fig. A1 and Fig. A2.

**A2 Leaf level: photosynthesis response curves**

Fig. A3a and Fig. A3b show the observed and modelled net $CO_2$ assimilation curves to $C_i$ and PAR.

**A3 Leaf level: procedure to calculate net $CO_2$ assimilation rate and leaf transpiration**

In this Sect. we detailed how we estimated the net assimilation rate and leaf transpiration based on observations. We have called these estimations post-processed observations because they combine observations of the closed chamber portable photosynthesis system (LiCOR 6400-XT) and of the in-canopy and above canopy sensors of T, PAR, specific humidity and $C_a$. Although the post-processed observations are not direct observations and they have certain limitations due to the assumptions made, we used them as a way of visualizing the essential diurnal variability of leaf fluxes. To study the leaf gas exchange in




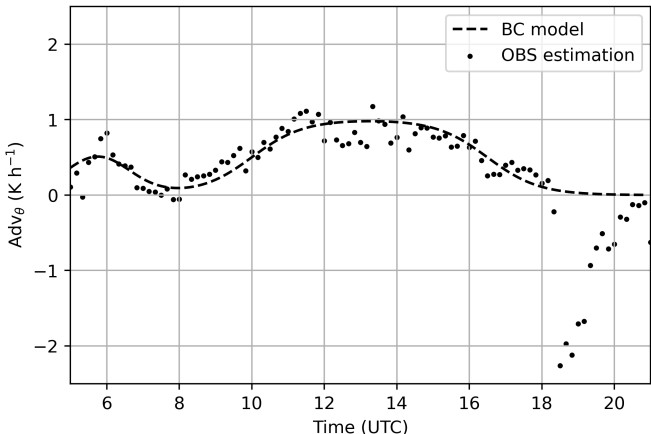

**Figure A1.** Temporal evolution of the temperature advection. Black dots indicate the estimations from the network of atmospheric weather stations whereas the dashed line shows the smoothed advection of temperature that was added as a boundary condition to Control experiment.

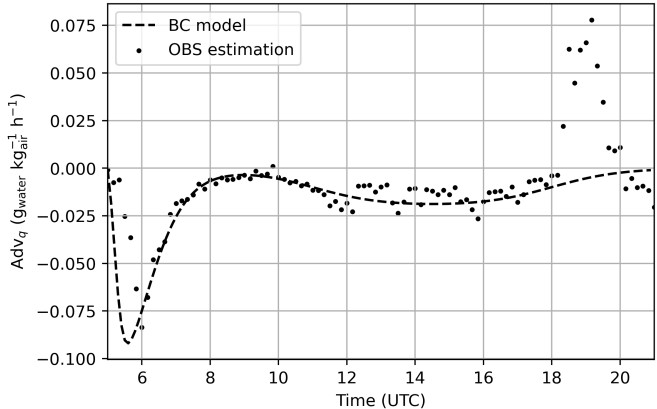

**Figure A2.** Same as A1 but for specific humidity advection.

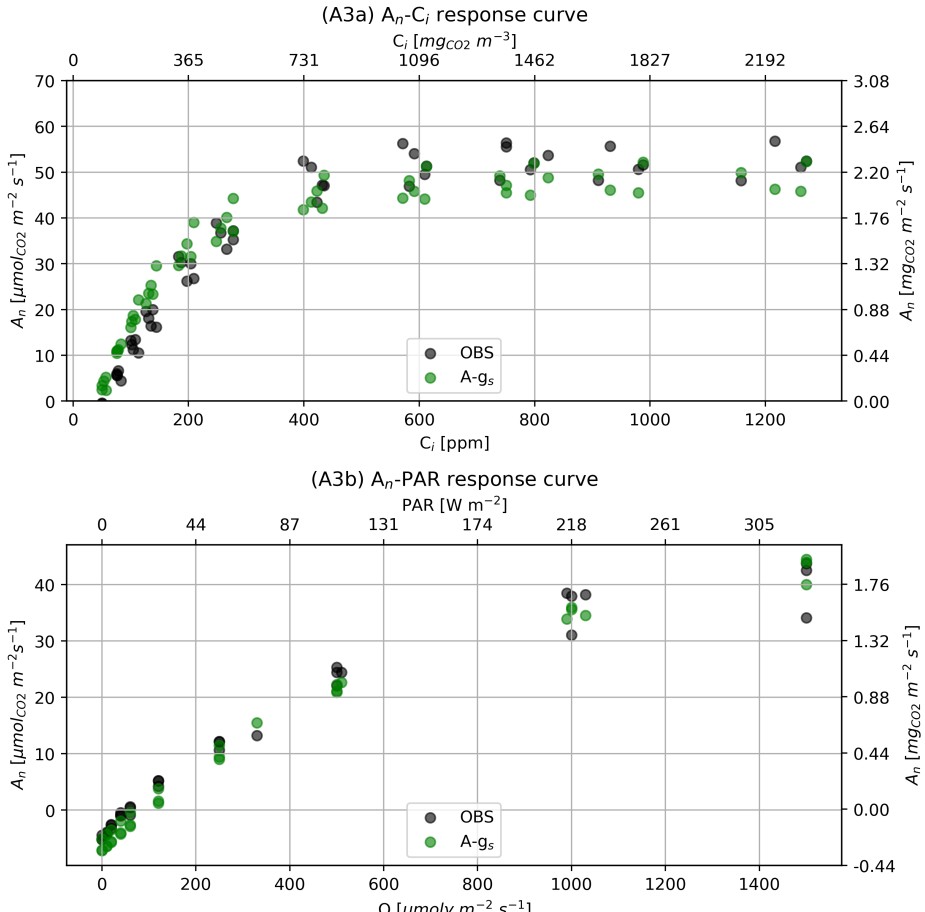

**Figure A3.** Response curves of net $CO_2$ assimilation rate to: (A3a) $C_i$ and (A3b) PAR or molar flux of photons (Q). Black dots are the observations whereas green dots are A-$g_s$ modelled results. A-$g_s$ predictions were made using the environmental conditions set in the leaf chamber (same temperature, radiation and $CO_2$ concentration) and the optimized parameters shown in Table 3.



greater detail another procedures should be used in combination. We do so, by complementing the observations with models of the leaf gas exchange.

The net assimilation rate and the leaf transpiration have been calculated using the following equations:

$$A_n = \frac{g_{s,w}}{\mu}\left(C_a - C_i\right) \tag{A2}$$

$$TR = g_{s,w}q_{sat}(T)\left(1 - RH\right)\frac{m_{air}}{m_{H2O}} \tag{A3}$$

where $\mu$ is the ratio of molecular diffusivities of water vapor and carbon dioxide, $m_{air}$ is the molecular mass of air, $m_{H2O}$ is the molecular mass of water and RH is the relative humidity of the air. T, RH and $C_a$ were taken from sensors inside the

canopy (T and RH at 0.105 m) and above the canopy ($C_a$ at 3 m). $C_i$ was calculated using the ratio of internal to external $CO_2$ concentration measured inside the closed chamber portable photosynthesis system (LiCOR 6400-XT) and the $C_a$ measured above the canopy.

*Author contributions.* Raquel González Armas carried out the numerical experiment and the corresponding analyses. Jordi Vilà-Guerau de Arellano, Hugo de Boer and Raquel González Armas developed the ideas that led to the study and discuss thoroughly the results.

Raquel González Armas developed and calculated the tendency equations. Hugo de Boer provided his expertise in the leaf gas exchange measurements and plant physiology, and he organized the leaf measurements plan for La Cendrosa in LIAISE field campaign providing also the needed equipement. Mary Rose Mangan and Oscar Hartogensis were in charge of the surface energy balance station at La Cendrosa and they also developed an initial CLASS simulation for the location which was then adapted for the current study. They also provided valuable discussions which shaped the research here presented. All authors read and approved the manuscript.

*Competing interests.* The authors declare that they have no conflict of interest.

*Acknowledgements.* The authors acknowledge the funding provided by the Dutch Research Council (NWO; under the project Cloud-Roots - Clouds rooted in a heterogeneous biosphere with project number OCENW.KLEIN.407). The authors thank the organizers, hosts and participants of the LIAISE campaign because thanks to them the development of this research was possible. In particular, we would like to acknowledge Martin Best, Joaquim Bellvert, Jennifer Brooke, Jan Polcher and Pere Quintana for their work on the LIAISE steering

committee, and Henk Snellen, Getachew Adnew, Marc Castellnou, Siluo Chen, Kevin van Diepen, Siluo Chen, Kim Faassen, Wouter Mol, Robbert Moonen, Ruben Schulte and Gijs Vis for their work on the LIAISE dutch team during the experiment. We would also like to reiterate our gratitude to Kevin van Diepen, Siluo Chen and Kim Faassen for contributing with the measurements of the leaf gas exchange during LIAISE field Campaign.





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
