# Peer review of "On the impact of canopy environmental variables on the diurnal dynamics of the leaf and canopy water and carbon dioxide exchange"

_EGUsphere, 2023_

## Referee Comment (RC2)

The paper uses the field fata on the alfalfa and modelling to study the $CO_2$ and water vapor exchange at leaf, canopy and boundary-layer scales. The observations carried out on one clear day were further perturbed by inducing a cloud passage, entrainment of dry air and advection of cold air by the model. In addition, the tendency equations were used to explain the revealed responses in the exchange rates.

The paper is generally very clearly written and brings an important insight by its original approach. I have only the following minor comments regarding the first part of the paper and Results and Discussion parts are clear in their message:

Lines 44-45: The scales are explained. Is there any certain horizontal larger scale for analyses, especially for advection?

Fig. 1: $TR_{leaf}$ and An should be explained in Fig. caption. Why two arrows in the middle picture are not arrows but just triangles? To which solid and dashed lines are referred to in the caption? They are not lines but arrows.

Are there some more reasons for selection of that one day, beside that it is cloudless?

Was the measured LAI the total all-sided or half-sided or the projected one?

Table 2: why the unit for the thermal diffusivity is missing?

Table 3: why the cuticular minimum conductance is unitless? Why is the $CO_2$ compensation concentration value multiplied by the density of the air?

Lines 277-278: I don't understand the meaning of "but we did not…present in observations.".

---

## Author Comment (AC1)

**Response to reviewer # 1 of the manuscript "On the impact of canopy environmental variables on the diurnal dynamics of the leaf and canopy water and carbon dioxide exchange" by González-Armas et al.**

First of all, we would like to thank the anonymous reviewer for his/her constructive assessment of the manuscript. We have considered all the points raised by the reviewer and adjusted the manuscript accordingly. For clarity, the specific comments of the reviewer are repeated in black font and then addressed by us in blue font. In *italics font and between quotation marks "",* we write how we plan to modify the manuscript. From that text, we use blue to indicate parts of the non-edited manuscript that will remain the same, in  we indicate parts that will be removed in the edited version and in green font new additions.

**Main comments**

1. The tendency equations represent a powerful tool to "make sense" of modelling output. Often in the literature, modelling outputs (or differences in modelling outputs) are not sufficiently explained and tendency equations might help in this regard. I was wondering if a framework could be developed to apply this concept to observations or at least to use observations to validate these tendencies in models. In the current study, observations are shown to demonstrate that the model can reproduce the diurnal dynamics in the observations. However, it would for example be even more insightful to test if the opposing $A_n$ responses at the leaf- and canopy-scale in the VPD-ENT experiment can be validated against field observations. These questions could also be discussed as part of the Discussion.

We acknowledge the relevance of the point raised by the reviewer and actually some of the authors of the manuscript has recently published a paper in which a tendency equation for the latent heat flux at canopy scale is calculated from observations for the same region (Mangan, Hartogensis, van Heerwaarden, & Vilà-Guerau de Arellano, 2023). In the current manuscript however, our aim was to introduce the tendencies of the leaf gas exchange and use them as a diagnostic tool to analyze the sensitivity of leaf fluxes to environmental variables and specific environmental processes (advection, entrainment, and a cloud passage). For this kind of systemic, process-based sensitivity study a couple model is needed. Although the approach of (Mangan, Hartogensis, van Heerwaarden, & Vilà-Guerau de Arellano, 2023) could in principle be adapted to the leaf gas exchange tendencies, the challenges related to data continuity and scatter would place this approach out of the scope of the manuscript. Still, we agree that the idea has merit, and we will add a discussion of the work (Mangan, Hartogensis, van Heerwaarden, & Vilà-Guerau de Arellano, 2023) , the value of it and the possibility to apply it to leaf gas exchange tendencies:

(starting from line 455)

*"Tendency equations, similar to the ones presented here, have been proposed in the past for leaf transpiration (Jarvis and McNaughton, 1986), evapotranspiration (van Heerwaarden et al., 2010) and net ecosystem exchange (Pedruzo-Bagazgoitia et al., 2017) but with substantial differences with respect to our study. Jarvis and McNaughton (1986) used a similar approach to investigate the dependency of transpiration on stomatal conductance for scales ranging $10-5$ m up to $105$ m. The approach was different because it was not intended to analyze temporal dynamics of the fluxes but to investigate the sensitivity of transpiration on stomatal conductance at different scales. Because of that, Jarvis and McNaughton (1986) used differential equations but not with respect to time. Additionally, the $CO_2$ fluxes were not investigated and to the author's knowledge, this is the first time that the tendencies have been calculated simultaneously for stomatal conductance, leaf transpiration and net assimilation rate providing a complete view*

*of the leaf gas exchange. On the other hand, van Heerwaarden et al. (2010) calculated tendency equations for the canopy evapotranspiration and Pedruzo-Bagazgoitia et al. (2017) for the gross primary productivity. Both of them applied the approach to investigate diurnal dynamics in realistic field conditions. The approach of van Heerwaarden et al. (2010) was based on Penman–Monteith equation combined with mixed-layer theory for CBL whereas Pedruzo-Bagazgoitia et al. (2017) was based on the up-scaled CO2 flux given by A-gs (eq. (3)). The main difference between those approaches and the one presented here is that we calculated the terms as a function of state primary variables whereas van Heerwaarden et al. (2010) and Pedruzo-Bagazgoitia et al. (2017) did it for intermediate variables. For example, a term of the equation proposed by Pedruzo-Bagazgoitia et al. (2017) contained the temporal derivative of Ci which may be more difficult to interpret and relate to environmental processes than changes in PAR, T, VPD and $C_a$. We acknowledge that for certain research questions it may be relevant to use a different subset of independent variables. However, the choice of another subset is also possible within the proposed framework. Eventually, we would like to comment on the possibility of combining tendency equations and observations. Whereas the previous cited research had calculated tendencies with coupled models, (Mangan, Hartogensis, van Heerwaarden, & Vilà-Guerau de Arellano, 2023) calculated the LE tendencies derived by (van Heerwaarden et al., 2010) and apply them to the coupled model CLASS but also to observations. By doing so they could explore whether the model and observations agreed that certain processes impact LE in the same manner. In principle, a similar approach could be developed for the tendencies introduced in the current manuscript. Although observations will introduce noise to the tendencies, a comparison between tendencies of the leaf gas exchange applied to model and observations could provide an additional tool to assess the performance of the A-$g_s$ scheme apart from directly comparing leaf fluxes."*

Regarding the last part of the comment that referred to the opposing responses of $A_n$ at leaf and canopy level for VPD-ENT experiment, we cannot explore this surprising result with observations during the campaign because this finding occurred for one idealized experiment (VPD-ENT). Additionally, it is important to note that we have been able to notice this unexpected result because we are comparing it against a control case, something that cannot be achieved with observations. We would also like to mention that the opposing responses are small in magnitude (Fig. 8 in the manuscript; less than 2% of increase/decrease during the daytime compared to Control), therefore it may be challenging to observe it in a field campaign. We will elaborate more on this matter in comment number 13.

2.  The authors mention the potential effects of diffuse and direct radiation on carbon and water exchange. I think this is an important issue and should be more discussed in the Discussion section. The authors could go further than just mentioning potential effects and discuss how for example cloud cover and the associated changes in direct and diffuse radiation could affect their results.

We appreciate the reviewer's remarks regarding the importance of not only the magnitude but also the characteristics of the light in influencing the carbon and water exchange dynamics. In response to this suggestion, we will revise the paragraph three of the discussion, splitting it in two parts. The second part now will include a detailed discussion on the impact that the partitioning of shortwave radiation into its direct and diffuse components has in our results. Furthermore, we plan to add two figures of observations of direct and diffuse components of radiation for two days during LIAISE campaign in the supplementary material. In the revised manuscript we also have incorporated relevant literature on the topic in the discussion section (Niyogi, et al., 2004) (Knohl & Baldocchi, 2008) (Pedruzo-Bagazgoitia, et al., 2017).

The edited discussion would be as follows:

(starting from line 446 which would be the start of the new paragraph)

"Based on the modelled leaf gas exchange, tendency equations were used to quantify the effect of the diurnal dynamics of the environmental drivers on the dynamics of the leaf gas exchange. In that regard, the tendency terms informed about the modelled leaf gas exchange and are bounded by the  assumptions of the same. ~~Some additions that could be included to the A-gs scheme and tendency equations, but were not in this research, are the distinction between direct and diffuse light (Gu et al. 2002; Pedruzo-Bagazgoitia et al. 2017; Durand et al. 2021) and the temporal adaptation of the stomata to instantaneous changes in environmental conditions (Sellers et al. 1996; Vico et al. 2011; Sikma et al. 2018). Sensitivities to these two additions may be important to generalize findings about cloud-vegetation interactions specially for fast radiative perturbations as those that have been observed and modelled in previous research (Kivalov and Fitzjarrald 2018 and Mol et al. 2023).~~ An addition that could be included to the A-$g_s$ scheme is the temporal adaptation of the stomata to instantaneous changes in environmental conditions (Sellers et al. 1996; Vico et al. 2011; Sikma et al. 2018). Adaptation of the stomata could be important specially for fast radiative perturbations as those that have been observed and modelled in previous research (Kivalov and Fitzjarrald 2018 and Mol et al. 2023). Another feature that was not accounted for in the numerical experiments was the partitioning of shortwave radiation between its direct and diffuse components. This partitioning can be important because diffuse light is considered to increase the vegetative canopy that is receiving illumination and therefore, it can increase the net $CO_2$ assimilated by the canopy as it has been previously reported (Niyogi, et al., 2004) (Knohl & Baldocchi, 2008). During the LIAISE field campaign, direct and diffuse components of shortwave radiation were measured at La Cendrosa (Fig. 1 and Fig. 2 in the supplementary material). During the studied day, the ratio of diffuse radiation to the net radiation was approximately 15 %, categorized as a low diffusive regime according to (Niyogi, et al., 2004). Therefore, we anticipate minimal impact of the partitioning of direct and diffuse for the Control experiment in which we based the largest part of our conclusions. For VPD-ENT and TEM-ADV numerical experiments, the partitioning of radiation remains consistent, suggesting a minor impact on our results. However, for the PAR-CLD numerical experiment, we acknowledge the substantial modification of partitioning that can be induced by clouds. In a cloudy day during the campaign, the ratio of diffuse radiation to net radiation oscillated between 35 % and 100 % (where values larger than 60 % are categorized as high diffusive according to (Niyogi, et al., 2004); Fig. 2 of supplementary material). Such cloud-induced changes in direct and diffuse partitioning could influence the $CO_2$ exchange, potentially leading to larger canopy $CO_2$ uptake in PAR-CLD compared to our results. For instance, (Pedruzo-Bagazgoitia, et al., 2017) researched the impact of direct and diffuse radiation using large eddy simulations and they found greater net canopy assimilation of $CO_2$ for cloud optical depths below 3 that could be as much as 9 % compared to clear sky values. We emphasize that the direct and diffuse partitioning is relevant to understand the vertical profiles of light within the canopy and therefore, it is considered when up-scaling fluxes from leaf level to canopy level. However, the leaf tendencies as they have been presented here could still be coupled to a model that accounts for direct and diffuse partitioning."

[Figure]

*Figure 1: Radiation components at La Cendrosa during the studied day. The inset figure depicts the ratio of diffuse radiation to net radiation ($R_n$). The blue shaded area depicts the low diffusive regime whereas the red shaded area depicts a high diffusive regime according to (Niyogi, et al., 2004).*

[Figure]

*Figure 2: Same than Figure 1 but for a cloudy day during the LIAISE field campaign.*

**Specific comments**

3. Line 41: Explain what the "phase lag" is.

The phase lag is a metric that accounts for the asymmetry between a flux or another land-surface variable and the incoming short-wave radiation (Renner et al, 2019b). As presented by Renner et al, (2019b) and Renner et al, (2021), the phase lag has units of time. For instance, a positive phase lag of LE indicates that LE is larger in the afternoon than in the morning with the magnitude of the phase lag quantifying its level of asymmetry. We will explain the term in the text:

"*When assessing the performance to reproduce heat fluxes, they considered both the magnitude and  a metric, called the phase lag (Renner et al, 2019b), that indicates the asymmetry between the heat fluxes and the incoming shortwave radiation. In their study they concluded that all LSMs showed a poor representation of the evaporative fraction and phase lag. The authors also highlighted the importance of systematic evaluations of the diurnal dynamics of the fluxes in order to improve the understanding and predictive capacity of the near-surface climate.*"

4. Line 44-52: The authors refer here to "surface" and "canopy". Is there a difference between these two definitions?

We will substitute "surface" by "canopy" in line 51. As pointed out by the reviewer, there was an inconsistency. In that paragraph, when we refer to surface, we refer to its most common meaning "the uppermost layer of something." Whereas in the sentence starting at line 51 we want to indicate that the atmospheric boundary layer interacts with the canopy, which is composed by vegetation elements (e.g., leaf surfaces) and by the soil surface.

5. Line 83: Why do the authors only focus on diurnal dynamics? The model could also be validated against multi-day simulations to explore day-to-day variations in meteorological conditions and their impacts.

We decided to focus on diurnal dynamics because modelled surface fluxes of carbon, water and energy are reported to show disagreement in their sub-diurnal variability with observations (Renner, et al., 2021). We introduced this topic in the first paragraph of the introduction (lines 33-43). Because of that gap in knowledge, we aimed at understanding the drivers of the diurnal variability of leaf fluxes which compose a large signal of the surface fluxes in vegetated areas. Although day-to-day variations could also be interesting to explore, we consider this analysis out of the scope of the current manuscript, specifically because the collection of leaf-level gas exchange is extremely time consuming and therefore it limited our data collection to two intense measurement days during which all other meteorological variables were also measured.

6. Figure 1: This is a great figure conceptualising their research approach.

We thank the reviewer for the positive comment.

7. Line 133: How was it tested if the ABL started to be a CBL?

We analyzed the vertical profiles measured by radiosondes released at the site. To explore the type of ABL we looked at potential temperature vertical profiles. The ABL was considered a CBL:

- o   The air temperature increased with height close to the surface.
- o   The air maintained a quasi-constant temperature in height from certain distance of the surface (approximately 50 m) up to the entrainment zone that was characterized by an inversion of temperature.

Looking at the first four radiosondes released at the site during the day, the one released at 6:56 LT (= 4:56 UTC ≈ 5:00 UTC) was the first that met the conditions, because of that, 7:00 LT (5:00 UTC) was the time at which CLASS was initialized.

[Figure]

Figure 3: Six first radiosondes released at La Cendrosa during 17/07/2021

We have re-written the sentence in the manuscript to clarify this question:

(line 132) "*Because with CLASS we describe a CBL, our analysis during the studied day was restricted from 5 UTC, when according to vertical profiles of potential temperature measured by radiosondes the ABL was a CBL. Our analysis was restricted until approximately 15:40 UTC, when our numerical experiment indicated a transition to non-convective conditions.*"

8.  Line 146: The authors mention here that the initial $CO_2$ jump was chosen to reproduce the diurnal variability in observed $CO_2$. Later, they validate the model against diurnal $CO_2$ dynamics. Would this not by default improve the model output? Is it then justified to compare the model $CO_2$ output to observations.

We acknowledge that there is certain circular reasoning in setting the initial value of the modelled $CO_2$ jump to reproduce the range of the diurnal variability in observed $CO_2$ and then comparing the model and observed $CO_2$ diurnal evolution. Because of the lack of information about atmospheric $CO_2$ concentrations, we had to assume the initial $CO_2$ jump. We chose an initial $CO_2$ jump that could reproduce the observed range of $CO_2$ concentrations during the day. The initial $CO_2$ jump influences the strength of the entrainment of air depleted of $CO_2$ from the free troposphere to the convective mixed-layer. The value mainly affects the magnitude of the morning drop of the mixed-layer $CO_2$. As mentioned by the reviewer, by choosing the initial $CO_2$ jump we influence the modelled $CO_2$ diurnal evolution which, by our choice, will resemble more closely to the EC observations. However, we only choose the initial value and although it influences the diurnal range of variability, it does not explain all the diurnal dynamics reproduced by the model. Because of that, model results and observations do have certain differences, such as the time when $CO_2$ dropped in the morning or the sharpness of the drop (Fig. 3 in the manuscript). We do not regard the comparison between observation and model results as a validation of the model but rather as a presentation of the temporal variability of the environmental drivers. We think that presenting the general temporal features of the environmental drivers (e.g. large $CO_2$ values in the morning and smaller values in the afternoon) is important to interpret the temporal dynamics of the fluxes. Additionally, we acknowledge that the available $CO_2$ observations may not be the best representation of the mixed-layer $CO_2$ values (points that are discussed in the paper between lines 429 and 436).

9. Line 186: How was the soil respiration derived from the chamber measurements? A reference could be sufficient here.

We determined the soil $CO_2$ efflux with the SRC-2 Soil Respiration Chamber and the EGM-5 Portable $CO_2$ analyzer as indicated in lines 186-187. The corresponding soil $CO_2$ efflux observations during the day are indicated as red dots in Fig 5a of the manuscript. How exactly the analyzer measures $CO_2$ concentration is explained in pages 11-12 of the operation manual of the EGM-5 Portable $CO_2$ analyzer (Systems, 2018). Details about the Soil Respiration chamber are given in page 15 of the same operation manual whereas how the analyzer works in combination with the chamber to measure soil $CO_2$ efflux is explained in pages 54-58 and in the Appendix A. To maintain consistency with our reporting of other instruments, we opted not to include the manual references. However, the operation manual for the EGM-5 Portable CO2 analyzer can be easily accessed online using the instrument name provided in the manuscript.

We have decided to use the term soil $CO_2$ efflux instead of soil respiration to refer to the observations measured by the device because it is more accurate to what the device provides (Maier, Schack-Kirchner, Hildebrand, & Schindler, 2011). Therefore, we will change the word in the text. Additionally, we will add in the text that we ensured that no alfalfa plant was inside the soil chamber. We had not indicated this in the previous version of the manuscript, but this action may be informative for a reader. The action was made to ensure that no above ground plant gas exchange would occur inside the soil respiration chamber.

Based on what have been said above we will modify the text starting at line 186 as follows:

"*We measured soil  $CO_2$ efflux with a SRC-2 Soil Respiration Chamber connected to a EGM-5 Portable CO2 Gas Analyzer. When measuring the soil $CO_2$ efflux, we ensure vegetation was not inside the chamber. In that way, no above-ground plant gas exchange would occur inside the chamber. We measured soil  $CO_2$ efflux at 7 times throughout the day (from 7:15 to 19:00 UTC) near the EC tower. Every time, three  or four soil  $CO_2$ efflux measurements were recorded. As a result, we obtain seven averaged values with its corresponding standard deviation.*"

In case the reviewer refers to how the soil respiration was estimated from the CLASS model (dashed line in Fig 5.), that is indicated in the line 176-177: *"Soil respiration is parameterized as a function of soil temperature and soil moisture. The surface and soil parameters used for the Control experiment are described in Table 2."*

10. Line 276: It could be helpful here to also show the actual observation in addition to the idealised modelling experiments.

Following the suggestion from the reviewer, we will refer to some figures in the manuscript that contain observations that inspired the experiments. For PAR-CLD and following the comment number 2, we decided to add a figure that shows the radiation budget during a cloudy day of the campaign in the supplementary material. This figure will be referred in the text. For TEM-ADV, there is already a figure in the appendix of the manuscript (Figure A2) that depicts the advection of cold air that was used in the numerical experiment. We will refer to that figure in the methods section. Finally, for VPD-ENT it has proved more difficult to show some observations because the estimations of the mixing ratio jump from the radiosondes are erratic and it is difficult to determine their exact value at the beginning of the simulations. Because of that, we have decided to provide a reference which carried out a similar sensitivity study with similar ranges for this parameter.

11. Line 308 & 309: I am not sure if "acquiring" and "excepting" are the right choice of words here.

We will modify "acquiring" by "reaching" and "excepting" by "except"

12. Section 3.1.3: Slope and intercept could be also shown for the model validation section.

We will add those values in the Supplementary Material (Figure 4 in this document) and we will refer to them in the results section of the leaf gas exchange.

[Figure]

*Figure 4: Predicted against observed leaf gas exchange variables.*

13. Line 384-397: It seems as if the different responses described in this paragraph are due to different model representation of photosynthesis. Is there any way to assess which representation is more accurate in this case?

In our opinion, the finding points out different responses at leaf and canopy level, which are likely to be related to the up-scaling formulation of $A_n$. As mentioned in response 1, it seems challenging to assess this with observations in a field campaign since the different responses

affect the $CO_2$ gas exchange to a small degree. Another approach to investigate this matter would be to test other modelling schemes and up-scaling approaches to explore whether this feature is consistent. We think this could be insightful, but we deem it to be outside of the scope of this manuscript.

14. Line 436: The authors could add more details on how the sensitivity analysis was conducted and what the results were.

The mentioned extra numerical experiment was carried out to quantify the impact of the mismatch between observed and modelled $C_a$ on our results. In the new experiment, called IMP-CO2, we forced modelled $C_a$ at 3 m to values similar to the $C_a$ measured by the eddy-covariance system at 3 m (Figure 5 in this document). We did not include this experiment in the study because in this numerical experiment the land-atmosphere model is not fully coupled unlike the other numerical experiments. In essence, in this experiment the $CO_2$ surface fluxes do not modify the atmospheric $CO_2$. IMP-CO2, which had lower $C_a$ than CONTROL, resulted in larger stomatal conductance values (approximately 5 % more than CONTROL averaged over the numerical experiment time), slightly lower leaf net $CO_2$ assimilation rate (approximately -2%) and slightly larger leaf transpiration (approximately 2%). However, the diurnal shape of the fluxes and the results of the tendencies remained similar between IMP-CO2 (Figure 7 in this document) and CONTROL (Figure 6 in the manuscript). The $C_a$ terms of the tendencies of IMP-CO2 numerical experiment had the same magnitude than the $C_a$ terms of CONTROL. The main difference on the $C_a$ terms was that they peaked earlier for IMP-CO2 than for CONTROL. However, the tendencies of IMP-CO2 led to the same conclusions of the study.

[Figure]

*Figure 5: Time series of Ca. Same as Fig. 3b of the manuscript but with grey lines depicting the Ca imposed in IMP-CO2 numerical experiment.*

[Figure]

*Figure 6: Same as Fig. 6 of the manuscript but for IMP-CO2 numerical experiment.*

Motivated by the comment, we will modify the discussion paragraph:

"*In this study, the coupled model CLASS could reproduce the observed diurnal variability of the environmental drivers for the studied day excepting for the variability of $C_a$ (Fig. 3b). Unlike for VPD and T, $C_a$ measurements were only available at 3 m and we did not have information about its vertical variability. CLASS model assumes that $C_a$ is well-mixed since the start of the numerical experiment. However, $C_a$ vertical profiles can depict strong vertical gradients during and after the morning transition from a stable ABL to an unstable and well-mixed ABL as it has been previously observed over grass (Casso-Torralba et al., 2008). As a consequence, the initial observed $C_a$ values may not be representative of the initial convective ABL. To explore the impact of the modelled and observed mismatch of the $CO_2$ diurnal evolution on our results, we performed an additional numerical experiment*  *in which modelled $C_a$ resembled closely to observed $C_a$. We have found that the leaf gas exchange tendencies retain its main features and they led to the same conclusions of the study.*"

---

## Author Comment (AC2)

**Response to reviewer # 2 of the manuscript "On the impact of canopy environmental variables on the diurnal dynamics of the leaf and canopy water and carbon dioxide exchange" by González-Armas et al.**

First of all, we would like to thank the anonymous reviewer for his/her constructive assessment of the manuscript. We have considered the points raised by the reviewer and adjusted the manuscript accordingly. For clarity, the specific comments of the reviewer are repeated in black font and then addressed by us in blue font. In *italics font and between quotation marks "",* we write how we plan to modify the manuscript. From that text, we use blue to indicate parts of the non-edited manuscript that will remain the same, in  we indicate parts that will be removed in the edited version and in green font new additions.

**Comments**

The paper uses the field data on the alfalfa and modelling to study the CO2 and water vapor exchange at leaf, canopy and boundary-layer scales. The observations carried out on one clear day were further perturbed by inducing a cloud passage, entrainment of dry air and advection of cold air by the model. In addition, the tendency equations were used to explain the revealed responses in the exchange rates.

The paper is generally very clearly written and brings an important insight by its original approach. I have only the following minor comments regarding the first part of the paper and Results and Discussion parts are clear in their message:

1. Lines 44-45: The scales are explained. Is there any certain horizontal larger scale for analyses, especially for advection?

   As pointed out, there are multiple scales that affect the state of the atmospheric boundary layer (ABL) at La Cendrosa and thereby also the surface fluxes. As mentioned by the reviewer, there are different regimes of advection of heat and moisture, for instance, sea breeze is regularly present over La Cendrosa. More details about salient weather characteristics of the region are presented by (Boone, et al., 2021). The area is also subject to thermal heterogeneity due to the presence of an irrigated region and a rain-fed region. We considered these larger scales into our coupled land-ABL model through the addition of some terms in the governing equations. Those terms are mainly advection of specific humidity and heat, the lapse rates of scalers of the free troposphere and the jumps of the scalars. Our approach was similar to that used by (Mangan, et al., 2023) to quantify and distinguish surface fluxes depending on the local, landscape or regional dominant scales. And in our present research we focus on the fluxes from the leaf up to the local scale. Additionally, in Fig. 1 of the manuscript, we introduce the main spatiotemporal scales involved in our study.

   To acknowledge this point, we have decided to include a sentence in the introduction:

   (In the introduction, lines 51-54)

   *"The ABL reacts to the dynamics of the surface and imposes forcings to it. Apart from surface processes, the ABL state also depends on non-local processes such as*

*entrainment of air from the free troposphere, advection of heat and moisture, and subsidence motions created by the influence of synoptic weather patterns."*

2. Fig. 1: TRleaf and An should be explained in Fig. caption. Why two arrows in the middle picture are not arrows but just triangles? To which solid and dashed lines are referred to in the caption? They are not lines but arrows.

We agree with the comments and consequently, we have edited the caption and figure. Firstly, we have introduced $TR_{leaf}$ and $A_n$ in the caption. Secondly, the two triangles have been replaced by arrows in the new edited figure. Lastly, we have rewritten the part of the caption related to the solid and dashed arrows since, thanks to the comments, we have realized that it was not clear. The edited caption reads now as:

*"Figure 1. Scheme of the three levels considered to study the exchange of water (represented in blue arrows) and carbon (represented in black arrows): (1) leaf level, (2) canopy level and (3) atmospheric boundary layer. The exchanges of water and $CO_2$ at leaf level are represented by the leaf transpiration ($TR_{leaf}$) and net $CO_2$ assimilation ($A_n$) respectively. At the atmospheric boundary layer level, several processes are included in the scheme such as advection of heat and moisture, and entrainment of air from the free troposphere. Advection and entrainment are indicated by solid arrows if they are contributing to higher concentrations of water or $CO_2$ in the boundary layer and dashed arrows if they contribute to lower concentrations.  In the scheme, we represent advection of moist and $CO_2$ enriched air and entrainment of drier and $CO_2$ depleted air from the free troposphere. ."*

3. Line 108, Are there some more reasons for selection of that one day, beside that it is cloudless?

The main reason to choose the 17/07/2021 as our studied day is because intensive leaf gas exchange measurements (shown in Fig. 4 of the submitted manuscript) were carried out. The other days, detailed information about the leaf gas exchange at La Cendrosa was missing.

4. Line 181, Was the measured LAI the total all-sided or half-sided or the projected one?

LAI was estimated with the ceptometer called ACCUPAR LP-80. The measurement of LAI was indirect because we do not measure leaf material by collecting samples. Instead, LAI was estimated based on an optical method. The instrument consists of two parts that measure PAR: (1) an external PAR sensor and (2) a probe containing 80 independent sensors, spaced 1 cm apart. The external PAR sensor was placed above the canopy to register the incident radiation whereas the probe was placed below the canopy. That measurements are used to estimate the canopy transmittance and to finally infer LAI with certain model that is explained on section 9.3 of the instrument manual (Decagon Devices, Inc., 2013).

The measured LAI with this method is an effective LAI because it is assumed that leaves are randomly distributed (Fang, Baret, Plummer, & Schaepman-Strub, 2019). Generally, LAI is defined as one half of the total green leaf area per unit horizontal ground surface area. That quantity is what the optical method aims to quantify. However, by inferring LAI from the PAR canopy transmittance, it is not distinguished the shade caused by a green leaf than that caused by any other vegetative tissue. Regarding classifying the measured LAI as half sided or projected LAI, (Barclay, 1998) classified the methods that infer LAI from canopy transmittance with similar instruments as projected LAI of inclined leaves.

5. Table 2: why the unit for the thermal diffusivity is missing?

It was a mistake. Now, we have added to Table 2 the units of thermal diffusivity of the skin layer which are: $W\ m^{-2}\ K^{-1}$.

6. Table 3: why the cuticular minimum conductance is unitless?

This was also a mistake. We have now added the units to Table 3 which are: $m\ s^{-1}$.

7. Why is the $CO_2$ compensation concentration value multiplied by the density of the air?

The multiplication was performed as a unit conversion to transform from ppm to a density. The density units are the ones used in the equations of the A-$g_s$ model (Ronda, De Bruin, & Holtslag, 2001). However, we think that indicating this conversion in Table 3 can be misleading. Because of that, we have opted to write the variable in ppm units since those are the most common units used in literature.

8. Lines 277-278: I don't understand the meaning of "but we did not...present in observations.".

In this piece, we wanted to explain two things. The first was that the radiative perturbation of the PAR-CLD sensitivity experiment was inspired on measurements during another day of the campaign were clouds present. In the revised manuscript, we have included information of this cloudy day in the Supplementary Material. The second point was that we did not represent sudden changes in radiation in PAR-CLD experiment. That sudden changes are also generally observed in cloudy conditions. However, we opted to not include them because in our model the stomata react instantaneously to the environmental variables. Because of that, fast changes in radiation may cause the tendencies to diverge and obscure their interpretation.

Unlike the first point, we think the second point regarding fast fluctuations fits better the discussion section rather than the methods section. Because of that, we have finally decided to remove it from methods and mention it only in discussion section (lines 451-452).

**Bibliography**

Boone, A., Bellvert, J., Best, M., Brooke, J., Canut-Rocafort, G., Cuxart, J., & Hartogensis, O. (2021). Updates on the international land surface interactions with the atmosphere over the Iberian semi-arid environment (LIAISE) field campaign. hal-03842003.

Mangan, M. R., Hartogensis, O., Boone, A., Branch, O., Canut, G., Cuxart, J., . . . Vilà-Guerau de Arellano, J. (2023). The surface-boundary layer connection across spatial scales of irrigation-driven thermal heterogeneity: An integrated data and modeling study. *Agricultural and Forest Meteorology, 335*, 109452.

Ronda, R. J., De Bruin, H. A., & Holtslag, A. A. (2001). Representation of the canopy conductance in modeling the surface energy budget for low vegetation. *Journal of Applied Meteorology and Climatology, 40*(8), 1431--1444.

Decagon Devices, Inc. (2013). *AccuPAR PAR/LAI Ceptometer Model LP-80. Operator's Manual.* Manual.

Fang, H., Baret, F., Plummer, S., & Schaepman-Strub, G. (2019). An Overview of Global Leaf Area Index (LAI): Methods, Products, Validation, and Applications. *Reviews of Geophysics, 57*(3), 739-799.

Barclay, H. J. (1998). Conversion of total leaf area to projected leaf area in lodgepole pine and Douglas-fir. *Tree Physiology, 18*(3), 185-193.

Fang, H., Baret, F., Plummer, S., & Schaepman-Strub, G. (2019). An overview of global leaf area index (LAI): Methods, products, validation, and applications. *Reviews of Geophysics, 57*(3), 739-799.

---

## Author Comment (AC3)

**Bibliography of the Response to reviewer # 1 of the manuscript:**

**"On the impact of canopy environmental variables on the diurnal dynamics of the leaf and canopy water and carbon dioxide exchange" by González-Armas et al.**

**Bibliography**

Maier, M., Schack-Kirchner, H., Hildebrand, E. E., & Schindler, D. (2011). Soil CO2 efflux vs. soil respiration: Implications for flux models. *Agricultural and forest meteorology, 151*(12), 1723-1730.

Systems, P. (2018). *EGM-5 Portable CO2 Gas Analyzer Operation Manual.* Amesbury: PP Systems.

Mangan, M. R., Hartogensis, O., van Heerwaarden, C., & Vilà-Guerau de Arellano, J. (2023). Evapotranspiration controls across spatial scales of heterogeneity. *Quarterly Journal of the Royal Meteorological Society, 149*(756), 2696-2718.

Niyogi, D., Chang, H. I., Saxena, V. K., Holt, T., Alapaty, K., Booker, F., & Xue, Y. (2004). Direct observations of the effects of aerosol loading on net ecosystem CO2 exchanges over different landscapes. *Geophysical Research Letters, 31*(20).

Knohl, A., & Baldocchi, D. D. (2008). Effects of diffuse radiation on canopy gas exchange processes in a forest ecosystem. *Journal of Geophysical Research: Biogeosciences, 113*(G2).

Pedruzo-Bagazgoitia, X., Ouwersloot, H. G., Sikma, M., Van Heerwaarden, C. C., Jacobs, C. M., & De Arellano, J. V. (2017). Direct and diffuse radiation in the shallow cumulus–vegetation system: Enhanced and decreased evapotranspiration regimes. *Journal of Hydrometeorology, 18*(6), 1731-1748.

Renner, M., Kleidon, A., Clark, M., Nijssen, B., Heidkamp, M., B. M., & Abramowitz, G. (2021). How well can land-surface models represent the diurnal cycle of turbulent heat fluxes? *Journal of Hydrometeorology, 22*(1), 77-94.